# FLOW-BASED CONFORMAL PREDICTION FOR MULTI-DIMENSIONAL TIME SERIES

**Junghwan Lee, Chen Xu & Yao Xie**
H. Milton Stewart School of Industrial and Systems Engineering
Georgia Institute of Technology
{jlee3541,cxu310}@gatech.edu, yao.xie@isye.gatech.edu

## ABSTRACT

Time series prediction underpins a broad range of downstream tasks across many scientific domains. Recent advances and increasing adoption of black-box machine learning models for time series prediction highlight the critical need for uncertainty quantification. While conformal prediction has gained attention as a reliable uncertainty quantification method, conformal prediction for time series faces two key challenges: (1) **leveraging correlations in observations and non-conformity scores to overcome the exchangeability assumption**, and (2) **constructing prediction sets for multi-dimensional outcomes**. To address these challenges, we propose a novel conformal prediction method for time series using flow with classifier-free guidance. We provide coverage guarantees by establishing exact non-asymptotic marginal coverage and a finite-sample bound on conditional coverage for the proposed method. Evaluations on real-world time series datasets demonstrate that our method constructs significantly smaller prediction sets than existing conformal prediction methods, maintaining target coverage.

## 1 INTRODUCTION

Uncertainty quantification has become essential in scientific fields where black-box machine learning models are widely deployed (Angelopoulos & Bates, 2021). Conformal prediction (CP) has emerged as a distribution-free framework for uncertainty quantification with coverage guarantees, ensuring prediction sets contain the true outcome at a specified confidence level (Shafer & Vovk, 2008; Vovk et al., 2005). By constructing uncertainty sets using non-conformity scores that quantify how atypical predictions are, CP generates reliable prediction sets that satisfy a specified confidence level.

Time series prediction aims to forecast future outcomes from past sequential observations of features and outcomes (Box et al., 2015), underpinning a broad range of downstream tasks. Recent advances in machine learning have led to the development of various foundation models designed for time series prediction (Kim et al., 2025; Miller et al., 2024; Wen et al., 2023). The growing adoption of such models for time series prediction highlights the pressing need for reliable uncertainty quantification. Although CP has been actively studied for reliable uncertainty quantification, most existing CP methods rely on the assumption of data exchangeability (Barber et al., 2023). The exchangeability assumption is frequently violated in time series data, where observations exhibit complex temporal dependencies that induce correlations in the non-conformity scores, thereby making the direct application of CP to time series prediction particularly challenging. An additional challenge is that modern time series data of-

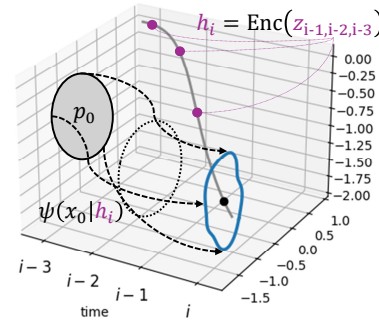

Figure 1: Our method adaptively constructs the prediction set using a flow transformation $\psi$ conditioned on guidance $h_i$, which encodes the past context.

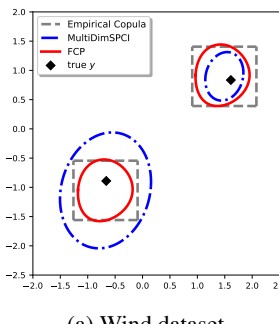 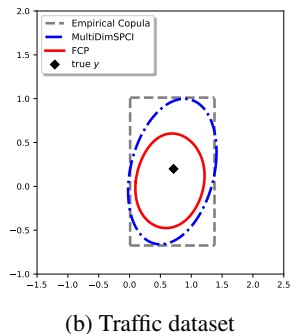 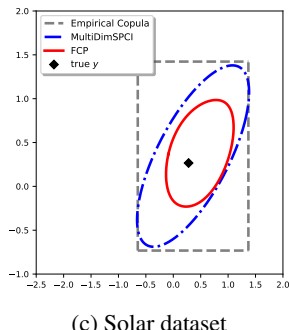

(a) Wind dataset  (b) Traffic dataset  (c) Solar dataset

Figure 2: Comparison of the prediction sets at a target coverage of 0.95, constructed by FCP (ours), MultiDimSPCI (Xu et al., 2024), and conformal prediction using empirical copula (Messoudi et al., 2021) on (a) wind, (b) traffic, and (c) solar datasets. Prediction sets are manually selected from the test set for visual clarity. Two prediction sets are shown for the wind dataset.

ten contain multi-dimensional outcomes. While CP methods for univariate outcomes are well-established, extending these methods to generate prediction sets for multi-dimensional outcomes is not straightforward and requires careful consideration in constructing prediction sets.

There has been substantial effort to extend CP beyond the exchangeability assumption. One line of research focuses on addressing distribution shifts in the data (Barber et al., 2023; Tibshirani et al., 2019). More recently, several works have developed CP methods for time series. For example, Xu & Xie (2021a) proposed a method to construct sequential prediction intervals for time series based on a bootstrap ensemble estimator, which were later extended to incorporate conditional quantile estimation in order to exploit correlations in non-conformity scores (Xu & Xie, 2023b). Auer et al. (2024) used modern Hopfield networks to capture temporal dependencies for sample reweighting and constructed prediction intervals based on these weights. Another line of work has proposed multi-step conformal prediction methods for time series, but these approaches assume access to multiple i.i.d. time series sequences (Stankeviciute et al., 2021; Sun & Yu, 2022), which may limit their applicability in practical settings. Despite these efforts, most of the existing methods remain limited to univariate outcomes.

Constructing prediction sets for multi-dimensional outcomes has been an active area of research. Early approaches used copulas (Messoudi et al., 2021) and ellipsoidal uncertainty sets (Henderson et al., 2024; Johnstone & Ndiaye, 2022; Messoudi et al., 2022), yielding hyper-rectangular and ellipsoidal prediction sets, respectively. Subsequent research has aimed to move beyond specific geometric shapes of prediction sets: Braun et al. (2025) formulated structured non-convex optimization to obtain minimum-volume sets; and Tumu et al. (2024) used convex templates for prediction sets. Recent works have focused on transporting multi-dimensional non-conformity scores to a reference distribution from which prediction sets can be constructed. For example, Klein et al. (2025) and Thurin et al. (2025) used Monge–Kantorovich ranks (Chernozhukov et al., 2017; Hallin et al., 2021) to map multi-dimensional non-conformity scores onto a reference distribution to construct prediction sets, by solving optimal transport problems. Fang et al. (2025) applied conditional normalizing flows to map multi-dimensional non-conformity scores to the source distribution and construct prediction sets using a calibration set with the source distribution.

Consequently, an effective CP method for time series prediction should address the two aforementioned challenges: overcoming the exchangeability assumption and constructing prediction sets for multi-dimensional outcomes. To the best of our knowledge, Xu et al. (2024) is the only work that seeks to jointly addresses both challenges, constructing ellipsoidal prediction sets by defining non-conformity scores as the radii of the sets and predicting these non-conformity scores conditionally.

In this work, we propose a novel conformal prediction method for time series prediction with multi-dimensional outcomes. Our method is designed to effectively address the two challenges by using flow with classifier-free guidance. Specifically, we use flow to model the distribution of prediction residuals and their transformations conditioned on past context, which is encoded by using Transformer (Vaswani et al., 2017). We define the non-conformity score as the Euclidean distance between the transformed prediction residual and the mean of a Gaussian source distribution of the

flow, which allows us to construct prediction sets at a desired confidence level. We provide theoretical coverage guarantees by establishing an exact non-asymptotic marginal coverage and a finite-sample bound on conditional coverage for the proposed method. Empirical evaluations on three real-world multi-dimensional time series datasets demonstrate that the proposed method constructs significantly smaller prediction sets while maintaining target coverage, outperforming existing baselines.

## 2 PROBLEM SETUP

We consider a sequence of observations $\{(x_i, y_i) : i = 1, 2, \ldots\}$, where $x_i \in \mathbb{R}^{d_x}$ represents $d_x$-dimensional feature, and $y_i \in \mathbb{R}^{d_y}$ represents $d_y$-dimensional continuous outcome. We assume that we have a base predictor $\hat{f}$ that provides a point prediction $\hat{y}_i$ for $y_i$, given by $\hat{y}_i = \hat{f}(x_{(i-k):i})$, where $k$ specifies the size of the past observation window. The base predictor $\hat{f}$ can be any black-box model and is not restricted to any specific constraints.

Suppose that the first $T$ examples, $\{(x_i, y_i)\}_{i=1}^{T}$, are used for training. Our goal is to sequentially construct a prediction set $\hat{C}_i(z_i, \alpha)$ for the next step, beginning at time $i = T + 1$. Here, $z_i$ denotes the features used to construct $\hat{C}_i$, and $\alpha \in [0, 1]$ denotes a pre-specified significance level. In the simplest setting, $z_i$ consists only of $x_i$, but it may also include past features, outcomes, or non-conformity scores. We aim to construct prediction sets that satisfy *marginal coverage*:

$$\mathbb{P}\left(y_i \in \hat{C}_i(z_i, \alpha)\right) \geq 1 - \alpha, \quad \forall i, \tag{1}$$

and ideally *conditional coverage*:

$$\mathbb{P}\left(y_i \in \hat{C}_i(z_i, \alpha) \mid z_i\right) \geq 1 - \alpha, \quad \forall i. \tag{2}$$

Although trivially large prediction sets can always satisfy the target coverage, they do not provide useful information for uncertainty quantification. Therefore, the meaningful objective is to construct efficient prediction sets—the prediction sets that are as small as possible while satisfying the target coverage (Vovk et al., 2005).

Throughout this paper, we distinguish between the indices $i$ and $t$ to avoid confusion: the subscript $i$ refers to the discrete time index of the sequence of observations, while the subscript $t$ is reserved to refer to continuous time in ODEs. We use uppercase letters (e.g., $X$) to denote random variables and lowercase letters (e.g., $x$) to denote their realizations.

## 3 METHOD

### 3.1 PRELIMINARY: GUIDED FLOW

We use $x$ as a generic variable in this section, distinct from the time series feature $x_i$ introduced in the problem setup. A flow is a time-dependent mapping $\psi : [0, 1] \times \mathbb{R}^d \to \mathbb{R}^d$ that push-forward a random variable $X_0 \in \mathbb{R}^d$ from a source distribution $p_0$ to $X_t \in \mathbb{R}^d$ from a time-dependent probability density (i.e., probability path) $p_t$ at time $t \in [0, 1]$ as follows:

$$([\psi_t]_* p_0)(x_t) = p_0(\psi_t^{-1}(x_t)) \left| \det \frac{\partial \psi_t^{-1}}{\partial x_t}(x_t) \right|, \tag{3}$$

where $*$ denotes the push-forward operator, $\det(\cdot)$ denotes the determinant, and $\psi_t(x) := \psi(t, x)$. Flow $\psi$ is defined by a vector field $u : [0, 1] \times \mathbb{R}^d \to \mathbb{R}^d$ through the following ordinary differential equation (ODE):

$$\frac{d}{dt}\psi_t(x_0) = u_t(\psi_t(x_0)), \qquad \text{(flow ODE)}$$

$$\psi_0(x_0) = x_0. \qquad \text{(initial condition)} \tag{4}$$

A guided flow $\psi_{t|h} : [0, 1] \times \mathbb{R}^d \times \mathbb{R}^{d_h} \to \mathbb{R}^d$ enables conditional generation by learning a mapping from a source distribution to a target conditional distribution, and is defined by a guided vector field

$u_{t|h} : [0,1] \times \mathbb{R}^d \times \mathbb{R}^{d_h} \to \mathbb{R}^d$ with the following ODE:

$$\frac{d}{dt}\psi_{t|h}(x_0 \mid h) = u_{t|h}\left(\psi_{t|h}(x_0 \mid h) \mid h\right), \qquad \text{(guided flow ODE)}$$

$$\psi_{t=0|h}(x_0 \mid h) = x_0, \qquad \text{(initial condition)} \tag{5}$$

where $h \in \mathbb{R}^{d_h}$ denotes the guidance. By appropriately designing a conditional probability path per sample $x_1$ interpolating $p_{0|x_1}(x \mid x_1) = p_0$ and $p_{1|x_1}(x \mid x_1) = \delta_{x_1}$, where $\delta_{x_1}$ denoting the Dirac delta distribution centered at $x_1$, we can obtain the marginal guided probability path:

$$p_{t|h}(x \mid h) = \int p_{t|x_1}(x \mid x_1)\, q(x_1 \mid h)\, dx_1, \tag{6}$$

which interpolates the source distribution $p_0$ and the target conditional distribution $q(x_1 \mid h)$. Given the conditional vector field $u_{t|x_1}$ that generates each conditional path $p_{t|x_1}$, the marginal guided vector field is obtained as:

$$u_{t|h}(x \mid h) = \int u_{t|x_1}(x \mid x_1)\frac{p_{t|x_1}(x \mid x_1)q(x_1 \mid h)}{p_{t|h}(x \mid h)}\, dx_1. \tag{7}$$

One can verify the marginal guided vector field generates the marginal guided probability path using the *continuity equation* (see Proposition A.1). Therefore, in order to learn the target conditional distribution, we parameterize the guided vector field with neural networks and train it to approximate the marginal guided vector field as closely as possible. A simple and effective way to train the guided vector field is through flow matching, which minimizes the mean-squared error between the conditional guided vector field and the parameterized guided vector field (Lipman et al., 2022):

$$\mathcal{L}_{\text{CFM}} = \mathbb{E}_{t,(x_1,h)}\left[\left\|u_{t|h}^\theta(x \mid h) - u_{t|x_1}(x \mid x_1)\right\|^2\right], \tag{8}$$

where $t \sim \text{Unif}[0,1]$, $(x_1, h) \sim q_{\text{data}}$, and $u_{t|h}^\theta$ is the guided vector field with parameters $\theta$.

We consider Gaussian conditional probability path defined as $p_{t|x_1}(x \mid x_1) = \mathcal{N}(x \mid \alpha_t x_1, \sigma_t^2 I_d)$, where $\mathcal{N}$ denotes the Gaussian kernel and $I_d \in \mathbb{R}^{d \times d}$ denotes the identity matrix. $\alpha_t, \sigma_t : [0,1] \to [0,1]$ are interpolating scheduler, which are smooth functions satisfying $\alpha_0, \sigma_1 = 0$, $\alpha_1, \sigma_0 = 1$, and $\frac{d}{dt}\alpha_t - \frac{d}{dt}\sigma_t > 0$ for $t \in (0,1)$. The guided vector field $u_{t|h}(x \mid h)$ can be reformulated as:

$$u_{t|h}(x \mid h) = u_t(x) + b_t \nabla_x \log p_{h|t}(h \mid x), \tag{9}$$

where $u_t(x)$ is unguided vector field, $b_t$ is a scalar constant regarding $\alpha_t$ and $\sigma_t$ (see Proposition A.2). Based on this reformulation, early approaches trained a separate classifier (Song et al., 2020) with a classifier scale $w > 1$ is beneficial in conditional generation in practice (Dhariwal & Nichol, 2021) :

$$\tilde{u}_{t|h}(x \mid h) = u_t(x) + wb_t \nabla_x \log p_{h|t}(h \mid x). \tag{10}$$

By using the identity $\nabla_x \log p_{t|h}(x \mid h) = \nabla_x \log p_t(x) + \nabla_x \log p_{h|t}(h \mid x)$, equation (10) can be equivalently rewritten as:

$$\tilde{u}_{t|h}(x \mid h) = (1-w)u_t(x) + wu_{t|h}(x \mid h). \tag{11}$$

Instead of modeling $u_t(x)$ and $u_t(x \mid h)$ separately, Ho & Salimans (2022) proposed using a single vector field to model both cases by assigning a null condition $h_\varnothing$ to represent the unguided vector field, which is known as classifier-free guidance (CFG):

$$\tilde{u}_{t|h}(x \mid h) = (1-w)u_{t|h}(x \mid h_\varnothing) + wu_{t|h}(x \mid h), \tag{12}$$

where $h_\varnothing$ denotes the guidance representing the unguided state of the vector field. The guided vector field can be trained using flow matching with the loss:

$$\mathcal{L}_{\text{CFM}}^{\text{CFG}} = \mathbb{E}_{t,\eta,(x_1,h)}\left[\left\|u_{t|h}^\theta(x \mid (1-\eta)h + \eta h_\varnothing) - u_{t|x_1}(x \mid x_1)\right\|^2\right], \tag{13}$$

where $\eta \sim \text{Bernoulli}(p_\varnothing)$ and $p_\varnothing$ denotes the probability of assigning $h_\varnothing$. The resulting guided vector field $\tilde{u}_{t|h}(x \mid h)$ in equation (12) enables conditional generation by solving the guided flow ODE and has been widely used in various tasks such as image generation (Esser et al., 2024) and video generation (Polyak et al., 2024).

---

**Algorithm 1:** Training Guided Flow using Flow Matching

---

**Input:** $p_\varnothing$, initialized $u_{t|h}^\theta$ and $\text{Enc}^\theta$

**while** *not converged* **do**

    $\hat{\epsilon}_i \leftarrow y_i - \hat{y}_i$                          ▷ obtain prediction residuals

    $h_i \leftarrow \text{Enc}^\theta(z_i)$                           ▷ obtain guidance

    $h_i \leftarrow h_\varnothing$ with probability $p_\varnothing$        ▷ assign unguided state with probability $p_\varnothing$

    $x_0 \sim p_0(x)$

    $t \sim \text{Unif}(0,1)$

    $x_t \leftarrow \alpha_t \hat{\epsilon} + \sigma_t x_0$

    $u_{t|\hat{\epsilon}} \leftarrow \frac{d}{dt}\alpha_t \hat{\epsilon}_i + \frac{d}{dt}\sigma_t x_0$

    Update with $\nabla_\theta \|u_{t|h}^\theta(x_t, h_i) - u_{t|\hat{\epsilon}}\|^2$           ▷ flow matching loss

**Output:** trained $u_{t|h}^\theta$ and $\text{Enc}^\theta$

---

## 3.2 Conformal Prediction for Time Series using Guided Flow

We use guided flow to learn a mapping from the source distribution to the distribution of prediction residuals $\hat{\epsilon} = y - \hat{y}$, conditioned on past context. The prediction set is then defined through this transformation using guided flow to achieve the target coverage. This construction effectively addresses the two aforementioned challenges in conformal prediction for time series. First, guided flow leverages guidance from the encoder derived from past context, enabling accurate estimation of the conditional distribution of $\hat{\epsilon}$ and mitigating the reliance on the exchangeability assumption. Second, since guided flow defines transformations between random variables in arbitrary dimensions, it enables the generation of prediction sets for any multi-dimensional outcomes. Figure 1 provides a visual illustration of the method. We describe the method in detail in this section.

**Guided flow design.** We use Gaussian probability path with interpolating scheduler $a_t = t$ and $\sigma_t = (1 - t)$. The source distribution is set to an isotropic Gaussian with zero mean and covariance scale $\gamma > 0$, i.e., $\mathcal{N}(0, \gamma I_{d_y})$. For each time index $i$, we construct $z_i$ by concatenating the past $w$ features and prediction residuals, and use an encoder to obtain a contextual representation $h_i = \text{Enc}(z_i)$. The guided vector field as defined in equation (12) uses $h_i$ as the guidance to model the conditional distribution of $\hat{\epsilon}_i$. In our method, we use Transformer as the encoder (Vaswani et al., 2017), though alternative sequence models such as recurrent neural networks (RNNs) are also applicable. The guided vector field is trained via flow matching as defined in equation (13), and the encoder is jointly trained with it. The overall training procedure is summarized in Algorithm 1.

**Prediction set.** The trained guided flow models the conditional distribution of the prediction residual by mapping from the source Gaussian distribution to residuals conditioned on the guidance. Since this transformation is bijective, we can define prediction sets for the residuals directly through the transformation. Let $\hat{e}_i(y) := \|\psi_{t=1|h}^{-1}(\hat{\epsilon} \mid h_i)\|$ be the Euclidean distance between the transformed residual and the origin, then the prediction set at significance level $\alpha$ can be defined as:

$$\hat{C}_i(z_i, \alpha) = \{y : \hat{e}_i(y) \leq r_{1-\alpha}\},\tag{14}$$

where $r_{1-\alpha}$ is the radius of the ball $\mathcal{B}_{1-\alpha}$ that contains $1 - \alpha$ probability mass. Since we use $\mathcal{N}(0, \gamma I_{d_y})$ as the source distribution, the radius $r_{1-\alpha}$ is given by $r_{1-\alpha} = \sqrt{\gamma}\chi_{d_y}^{-1}(1-\alpha)$, where $\chi_{d_y}^{-1}$ denotes the inverse cumulative distribution function (CDF) of the chi distribution with $d_y$ degrees of freedom. Intuitively, the prediction set is obtained by taking the ball that contains the same amount of probability mass as the target coverage and transforming it to the prediction set for the residual using the guided flow. Although this construction directly uses $\hat{e}(y)$ to construct the prediction set, $\hat{e}(y)$ is computed from the transformed residual and therefore serves as a proxy non-conformity score, consistent with treating residuals as non-conformity scores.

Since the prediction set is obtained through the transformation using guided flow, it can take on flexible shapes without being constrained to follow any fixed geometric form, such as convex or ellipsoidal sets. We believe this enables the guided flow to generate smaller prediction sets that are better aligned with the data. Although the prediction sets do not have any fixed geometric shape, some useful topological properties can still be inferred. In particular, Theorem A.4 and A.5 ensure

that the prediction sets are closed and connected. Figure 2 shows prediction sets in $\mathbb{R}^2$ generated by our proposed method alongside two other methods that produce hyper-rectangular prediction sets (Messoudi et al., 2021) and ellipsoidal prediction sets (Xu et al., 2024). The figure visually demonstrates the flexible shapes of the prediction sets constructed by our proposed method.

The size of the prediction set is computed as:

$$\int_{\mathcal{B}_\alpha} \left| \det \left( J_{\psi_{t=1|h}}(x \mid h) \right) \right| \, dx \approx \text{Size}(\mathcal{B}_\alpha) \frac{1}{N} \sum_{j=1}^{N} \left| \det(J_{\psi_{t=1|h}}(x_j \mid h)) \right|, \qquad (15)$$

where $\psi_1$ represents the flow transformation from $t = 0$ to $t = 1$, and $J_{\psi_1}(x \mid h)$ denotes the Jacobian of $\psi_1$ at $x \in \mathcal{B}_\alpha$ conditioned on $h$. The right-hand side provides a Monte Carlo approximation, where $x_j$ are i.i.d. samples drawn from $\mathcal{B}_\alpha$ and $N$ is the number of samples. However, directly computing $\det \left( J_{\psi_1}(x \mid h) \right)$ is computationally expensive, as it requires solving the guided flow ODE and evaluating the full Jacobian. Instead, we compute the log-determinant of the Jacobian by solving the following ODE:

$$\frac{d}{dt} \log | \det J_{\psi_{t|h}}(x \mid h)| = \text{div} \left( u_{t|h}(\psi_{t|h}(x \mid h) \mid h) \right), \qquad \text{(Jacobian ODE)}$$
$$\log \left| \det \left( J_{\psi_{t=0|h}}(x \mid h) \right) \right| = 0, \qquad \text{(initial condition)} \qquad (16)$$

where $\text{div}(\cdot)$ denotes the divergence operator. A detailed derivation is provided in Proposition A.3. The accuracy of the prediction set size estimate depends on the Monte Carlo approximation. Purely random sampling from $\mathcal{B}_\alpha$ may introduce bias due to uneven coverage of the sampling space, and a small sample size $N$ can result in high variance. To reduce sampling bias, we use quasi-Monte Carlo sampling based on Sobol sequences (Sobol, 1967; Owen, 2023), which provides more uniform sampling from $\mathcal{B}_\alpha$. To control variance from finite sampling, we monitor the relative error in terms of sample size $N$. Additional implementation details are provided in the experiments section.

## 4 THEORY

In this section, we present exact non-asymptotic marginal coverage and a finite-sample bound on conditional coverage. We assume that $y_i \in \mathbb{R}^{d_y}$ is generated from an unknown true function $f$ with additive noise $\epsilon_i \in \mathbb{R}^{d_y}$ according to $y_i = f(x_{(i-k):i}) + \epsilon_i$. Proofs are presented in Appendix A.

### 4.1 MARGINAL COVERAGE

We first establish that prediction sets generated by our method achieve exact non-asymptotic marginal coverage. This result follows from a fundamental property of flow: probability mass preservation under push-forward operations. Lemma 4.3 formalizes this property and suffices to prove the exact non-asymptotic marginal coverage stated in Proposition 4.4.

**Assumption 4.1** (Flow existence and uniqueness). The guided vector field $u_t(x \mid h)$ is continuously differentiable and Lipschitz continuous in $x$ for all $t$ and $h$. That is, there exists a constant $L_u > 0$ such that

$$\|u_t(x \mid h) - u_t(x' \mid h)\| \le L_u \|x - x'\|, \quad \forall t, h, x, x'. \qquad (17)$$

*Remark* 4.2. Assumption 4.1 ensures the existence and uniqueness of solutions of the guided flow ODE. In practice, the guided vector field can be modeled using neural network architectures that satisfy this assumption, such as multi-layer perceptrons (MLP) with smooth activation functions.

**Lemma 4.3** (Probability mass preserving property of flows). *Let $X \sim p_X$ be a continuous random variable on $\mathbb{R}^d$, and let $\psi : \mathbb{R}^d \to \mathbb{R}^d$ be a $C^1$ diffeomorphism. Define $Y := \psi(X)$ with density $p_Y$ given by the push-forward of $p_X$ under $\psi$. Then, for any measurable set $\mathcal{A} \subset \mathbb{R}^d$, the transformed set $\mathcal{A}' := \psi(\mathcal{A})$ satisfies:*

$$\mathbb{P}(X \in \mathcal{A}) = \mathbb{P}(Y \in \mathcal{A}') \qquad (18)$$

**Proposition 4.4** (Marginal coverage). *Let $\alpha \in (0, 1)$ be a pre-specified significance level. Under Assumption 4.1, suppose the guided flow provides a sufficiently accurate approximation of the target distribution from the source distribution. If the ball $\mathcal{B}_{1-\alpha}$ defining the prediction set in equation (14) has probability mass $1 - \alpha$, then the prediction set achieves exact marginal coverage of $1 - \alpha$.*

### 4.2 CONDITIONAL COVERAGE

We next establish a finite-sample bound on conditional coverage. We define the non-conformity score based on the prediction residual as $\hat{e}_i = ||\psi^{-1}(\hat{\epsilon}_i \mid h_i)||$, and the non-conformity score based on the true noise as $e_i = ||\psi^{-1}(\epsilon_i \mid h_i)||$. The guided flow $\psi$ is trained on the training set until convergence and then fixed for computing $e$ and $\hat{e}$. The empirical CDF of $\hat{e}$ and $e$ are defined as:

$$\hat{F}_{T+1}(u) = \frac{1}{T} \sum_{i=1}^{T} \mathbb{1}\{\hat{e}_i \le u\}, \quad \tilde{F}_{T+1}(u) = \frac{1}{T} \sum_{i=1}^{T} \mathbb{1}\{e_i \le u\}. \tag{19}$$

We denote $F_e(u) = \mathbb{P}(e \le u)$ as the CDF of the true non-conformity scores. Since the source distribution of the guided flow in our method is set to be identical across time, the marginal distribution for $e_i$ can be considered to be identical for all $i$. However, while the marginal distribution of $e_i$ is identical for all $i$, they may exhibit dependence through $h_i$. Therefore, we consider two settings: (1) when $\{e_i\}_{i=1}^{T+1}$ are i.i.d., and (2) when $\{e_i\}_{i=1}^{T+1}$ are stationary and strongly mixing. We first establish a finite-sample bound on conditional coverage under the assumption of i.i.d. non-conformity scores.

**Assumption 4.5** (i.i.d. non-conformity scores). The true non-conformity scores $\{e_i\}_{i=1}^{T}$ are i.i.d.

**Assumption 4.6** (Bi-Lipschitz flow). We assume that the guided flow $\psi_t(x \mid h)$ is bi-Lipschitz continuous in $x$ for all $t$ and $h$. That is, there exist constants $L_\psi > 0$ and $L_{\psi^{-1}} > 0$, such that

$$\|\psi_t(x \mid h) - \psi_t(x' \mid h)\| \le L_\psi \|x - x'\|, \quad \forall t, h, x, x', \tag{20}$$

and

$$\|\psi_t^{-1}(x \mid h) - \psi_t^{-1}(x' \mid h)\| \le L_{\psi^{-1}} \|x - x'\|, \quad \forall t, h, x, x'. \tag{21}$$

*Remark* 4.7. Lemma A.7 shows that bi-Lipschitz guided vector field results in bi-Lipschitz guided flow. Consequently, we can model the guided vector field to satisfy this assumption. For example, one may use invertible Residual Networks (iResNet) (Behrmann et al., 2019; Chen et al., 2019) with smooth activation functions.

**Assumption 4.8** (Lipschitz continuous of the CDF of true non-conformity scores). $F_e(u)$ is Lipschitz continuous with Lipschitz constant $L_{T+1} > 0$, and that $F_e$ is strictly increasing in $u$.

**Assumption 4.9** (Estimation quality). Define $\Delta_i = \hat{\epsilon}_i - \epsilon_i$. There exists a sequence $\{\delta_T\}_{T \ge 1}$ such that

$$\frac{1}{T} \sum_{i=1}^{T} \|\Delta_i\|^2 \le \delta_T^2, \quad \|\Delta_{T+1}\| \le \delta_T. \tag{22}$$

As a result of Lemma A.9 and A.14, Theorem 4.10 establishes the finite-sample bound for conditional coverage under i.i.d. non-conformity scores.

**Theorem 4.10** (Conditional coverage bound under i.i.d. non-conformity scores). *Under Assumption 4.5, 4.6, 4.8, and 4.9, suppose the guided flow provides a sufficiently accurate approximation of the target distribution. With probability $1 - \delta$, we have:*

$$\left| \mathbb{P}(Y_{T+1} \in \widehat{C}_{T+1}^\alpha \mid Z_{T+1} = z_{T+1}) - (1 - \alpha) \right|$$
$$\le 12\sqrt{\frac{\log(16T)}{T}} + 4\left(L_{T+1} + \frac{1}{2}\right)(2C + \delta_T). \tag{23}$$

Next, we present a finite-sample bound on conditional coverage under the assumption of stationary and strongly mixing non-conformity scores.

**Assumption 4.11** (Strictly stationary and strongly mixing non-conformity scores). $\{e_i\}_{i=1}^{T}$ are strictly stationary and strongly mixing, with mixing coefficients satisfying $0 < \sum_{k>0} \alpha(k) < M < \infty$.

**Corollary 4.12** (Conditional coverage bound under stationary and strongly mixing non-conformity scores). *Under Assumption 4.6, 4.8, 4.9, and 4.11, suppose the guided flow provides a sufficiently accurate approximation of the target distribution. With probability $1 - \delta$, we have:*

$$\left| \mathbb{P}(Y_{T+1} \in \widehat{C}_{T+1}^\alpha \mid Z_{T+1} = z_{T+1}) - (1 - \alpha) \right|$$
$$\le 12\frac{(\frac{M}{2})^{1/3}(\log T)^{2/3}}{T^{1/3}} + 4\left(L_{T+1} + \frac{1}{2}\right)(2C + \delta_T). \tag{24}$$

The bounds in Theorem 4.10 and Corollary 4.12 depend on the sample size $T$ and the estimation error $\delta_T$. Both bounds converge to $1 - \alpha$ as $T \to \infty$, provided that $\delta_T = \mathcal{O}(T^{-a})$ for some $a > 0$. Intuitively, this implies the conditional coverage is guaranteed with sufficiently large training data and an accurate base predictor $\hat{f}$. The condition on $\delta_T$ can be satisfied by a broad class of estimators. For example, sieve estimators based on general neural networks achieve $\delta_T = o_p(T^{-1/4})$ when $f$ is sufficiently smooth (Chen & White, 1999). The Lasso estimator and Dantzig selector achieve $\delta_T = o_p(T^{-1/2})$ when $f$ is a sparse high-dimensional linear model (Bickel et al., 2009).

## 5 EXPERIMENTS

For notational convenience, we refer to our method as FCP, which stands for Flow-based Conformal Prediction. The guided vector field is modeled by an MLP with Softplus activations. We concatenate the guidance and time with the input for the MLP. dopri5 (Dormand & Prince, 1980) at absolute and relative tolerances of $10^{-5}$ is used to solve ODEs. A grid search is conducted to select the optimal hyperparameters for FCP. To determine an appropriate sample size $N$, we compute the relative standard error (SE) of the Jacobian determinants of $\psi$, defined as $\text{SE}\left(\{\det J_\psi(x_j \mid h)\}_{j=1}^N\right) / \text{Avg}\left(\{\det J_\psi(x_j \mid h)\}_{j=1}^N\right)$, then choose the smallest $N$ such that the average relative SE across all $h$ falls below 0.01. Additional details on hyperparameter selection and the choice of $N$ are provided in Appendix B.

**Baselines.** We evaluate FCP against several conformal prediction methods: MultiDimSPCI (Xu et al., 2024), OT-CP (Thurin et al., 2025), CONTRA (Fang et al., 2025), conformal prediction using local ellipsoids (Messoudi et al., 2022), CopulaCPTS (Sun & Yu, 2022), and conformal prediction using empirical and Gaussian copulas (Messoudi et al., 2021). We also include two widely used probabilistic time series forecasting methods as baselines: Temporal Fusion Transformer (TFT) (Lim et al., 2021) and DeepAR (Salinas et al., 2020). Although TFT and DeepAR are originally developed for time series with univariate outcomes, we adapt them to our multi-dimensional setting by constructing independent copulas using the predicted intervals for each output dimension. Further details on the baseline methods and their experimental setup are provided in Appendix B.

**Datasets and base predictor.** We evaluate FCP and baselines on three real-world time series datasets: wind, traffic, and solar datasets. Additional details on the datasets are provided in Appendix C. For the wind and traffic datasets, we randomly select $d_y \in \{2, 4, 8\}$ locations to construct five sequences of $d_y$-dimensional time series. For the solar dataset, we use $d_y \in \{2, 4\}$ and similarly construct five sequences. Base predictor $\hat{f}$ is required to provide a point prediction $\hat{y}$. We use two types of base predictors: (1) leave-one-out (LOO) bootstrap ensemble of 15 multivariate linear regressors, and (2) recurrent neural network (RNN) with long short-term memory (LSTM) units (Hochreiter & Schmidhuber, 1997). Since the RNN base predictor requires part of the sequence for training, whereas the LOO bootstrap predictor can leverage the full sequence, the effective sequence length available for evaluation varies by base predictor. The base predictor is trained independently for each of the five constructed sequences. For the RNN base predictor, the first 50% of each sequence was allocated for training, and predictions were made for the remaining 50%, which served as the evaluation sequence. Within this evaluation sequence, the first 80% was used as a training set, and the final 20% was evenly divided into validation and test sets. Since FCP does not require a calibration set to construct prediction sets, the validation set was used for model selection during training. To ensure fair evaluation in terms of data utilization, we combined the training and validation sets into a single calibration set for methods that require a calibration set.

**Evaluation metrics.** Efficient prediction sets are those that are as small as possible while satisfying the desired coverage. Therefore, we use two evaluation metrics: empirical coverage and the average prediction set size. The empirical coverage at a target confidence level $\alpha$ is defined as:

$$\frac{1}{|\mathcal{D}_{\text{test}}|} \sum_{\{z_i, y_i\} \in \mathcal{D}_{\text{test}}} \mathbb{1}\left(y_i \in \hat{C}_i(z_i, \alpha)\right), \tag{25}$$

where $\mathcal{D}_{\text{test}}$ denotes the test set. The average prediction set size is computed by averaging the sizes of $\hat{C}_i$ over the test set, with the specific definition of the set size depending on each method.

Table 1: Average empirical coverage and prediction sets size obtained by FCP and all baselines on three real-world datasets, evaluated under different base predictors and varying outcome dimensions $d_y$. Reported values represent the average and standard deviation over five independent experiments conducted on five constructed sequences. The target confidence level is set to 0.95. Results with average empirical coverage below the target confidence level are grayed out, and the smallest prediction set sizes, excluding the grayed-out results, are highlighted in bold.

| Dataset | Base Predictor | Method | $d_y = 2$ Coverage | $d_y = 2$ Size | $d_y = 4$ Coverage | $d_y = 4$ Size | $d_y = 8$ Coverage | $d_y = 8$ Size |
|---|---|---|---|---|---|---|---|---|
| **Wind** | LOO Bootstrap | FCP | $0.951_{\pm.018}$ | $\mathbf{0.88}_{\pm.089}$ | $0.953_{\pm.006}$ | $\mathbf{3.43}_{\pm1.37}$ | $0.956_{\pm.010}$ | $19.4_{\pm10.2}$ |
| | | MultiDimSPCI | $0.953_{\pm.016}$ | $1.31_{\pm.524}$ | $0.956_{\pm.018}$ | $6.39_{\pm3.90}$ | $0.951_{\pm.024}$ | $205.5_{\pm161.5}$ |
| | | CopulaCPTS | $1.0_{\pm.000}$ | $22.3_{\pm19.0}$ | $1.0_{\pm.000}$ | $611.3_{\pm484.7}$ | $1.0_{\pm.000}$ | $3.50\times10^5_{\pm3.73\times10^5}$ |
| | | OT-CP | $0.964_{\pm.015}$ | $2.71_{\pm1.54}$ | $0.958_{\pm.015}$ | $42.3_{\pm38.4}$ | $0.927_{\pm.027}$ | $1.28\times10^3_{\pm713}$ |
| | | CONTRA | $0.979_{\pm.024}$ | $32.9_{\pm25.8}$ | $1.000_{\pm.000}$ | $7.89\times10^5_{\pm1.49\times10^6}$ | $0.994_{\pm.006}$ | $5.88\times10^{11}_{\pm1.16\times10^{12}}$ |
| | | Local Ellipsoid | $0.964_{\pm.015}$ | $1.38_{\pm.419}$ | $0.971_{\pm.013}$ | $8.63_{\pm5.90}$ | $0.974_{\pm.011}$ | $394.9_{\pm522.4}$ |
| | | Empirical Copula | $0.951_{\pm.013}$ | $1.22_{\pm.316}$ | $0.958_{\pm.019}$ | $4.94_{\pm2.57}$ | $0.948_{\pm.012}$ | $77.4_{\pm26.1}$ |
| | | Gaussian Copula | $0.945_{\pm.017}$ | $1.17_{\pm.289}$ | $0.958_{\pm.019}$ | $5.11_{\pm2.40}$ | $0.948_{\pm.012}$ | $77.4_{\pm26.1}$ |
| | | TFT | $0.723_{\pm.172}$ | $1.34_{\pm.588}$ | $0.515_{\pm.174}$ | $4.26_{\pm3.52}$ | $0.187_{\pm.126}$ | $6.75_{\pm3.19}$ |
| | | DeepAR | $0.909_{\pm.036}$ | $1.32_{\pm.445}$ | $0.672_{\pm.130}$ | $4.84_{\pm3.86}$ | $0.320_{\pm.160}$ | $52.8_{\pm64.5}$ |
| | LSTM | FCP | $0.952_{\pm.054}$ | $\mathbf{1.18}_{\pm.215}$ | $0.957_{\pm.022}$ | $\mathbf{10.8}_{\pm1.05}$ | $0.953_{\pm.056}$ | $\mathbf{2.48\times10^3}_{\pm669}$ |
| | | MultiDimSPCI | $0.974_{\pm.009}$ | $3.79_{\pm1.71}$ | $0.926_{\pm.045}$ | $63.9_{\pm58.4}$ | $0.896_{\pm.035}$ | $5.53\times10^3_{\pm6.31\times10^3}$ |
| | | CopulaCPTS | $1.0_{\pm.000}$ | $45.7_{\pm45.4}$ | $1.0_{\pm.000}$ | $4.82\times10^3_{\pm3.73\times10^3}$ | $1.0_{\pm.000}$ | $2.83\times10^7_{\pm3.28\times10^7}$ |
| | | OT-CP | $0.970_{\pm.033}$ | $9.13_{\pm4.88}$ | $0.939_{\pm.052}$ | $212.3_{\pm124.5}$ | $0.943_{\pm.053}$ | $8.39\times10^4_{\pm4.68\times10^4}$ |
| | | CONTRA | $0.826_{\pm.201}$ | $0.317_{\pm.222}$ | $0.804_{\pm.178}$ | $0.192_{\pm.124}$ | $0.761_{\pm.205}$ | $25.0_{\pm35.2}$ |
| | | Local Ellipsoid | $0.978_{\pm.043}$ | $10.5_{\pm6.97}$ | $1.0_{\pm.000}$ | $354.4_{\pm406.8}$ | $1.0_{\pm.000}$ | $2.63\times10^5_{\pm2.70\times10^5}$ |
| | | Empirical Copula | $0.983_{\pm.035}$ | $14.2_{\pm8.19}$ | $1.0_{\pm.000}$ | $494.5_{\pm196.1}$ | $1.0_{\pm.000}$ | $4.46\times10^5_{\pm9.82\times10^4}$ |
| | | Gaussian Copula | $0.983_{\pm.035}$ | $14.1_{\pm8.18}$ | $1.0_{\pm.000}$ | $499.1_{\pm189.5}$ | $1.0_{\pm.000}$ | $5.24\times10^5_{\pm1.89\times10^5}$ |
| | | TFT | $0.550_{\pm.321}$ | $1.90_{\pm.695}$ | $0.395_{\pm.195}$ | $3.93_{\pm2.01}$ | $0.136_{\pm.189}$ | $23.7_{\pm34.8}$ |
| | | DeepAR | $0.786_{\pm.065}$ | $1.69_{\pm.489}$ | $0.305_{\pm.258}$ | $9.88_{\pm10.1}$ | $0.00_{\pm.000}$ | $22.8_{\pm32.6}$ |
| **Traffic** | LOO Bootstrap | FCP | $0.957_{\pm.014}$ | $\mathbf{0.915}_{\pm.119}$ | $0.953_{\pm.009}$ | $\mathbf{1.06}_{\pm.431}$ | $0.965_{\pm.015}$ | $\mathbf{1.53}_{\pm.161}$ |
| | | MultiDimSPCI | $0.963_{\pm.008}$ | $1.58_{\pm.446}$ | $0.968_{\pm.006}$ | $2.62_{\pm.908}$ | $0.971_{\pm.004}$ | $10.7_{\pm4.60}$ |
| | | CopulaCPTS | $1.000_{\pm.000}$ | $21.6_{\pm16.3}$ | $1.000_{\pm.000}$ | $645.8_{\pm645.5}$ | $1.000_{\pm.000}$ | $3.18\times10^5_{\pm4.80\times10^5}$ |
| | | OT-CP | $0.966_{\pm.008}$ | $2.03_{\pm.685}$ | $0.963_{\pm.007}$ | $32.0_{\pm20.0}$ | $0.954_{\pm.007}$ | $3.90\times10^3_{\pm1.22\times10^3}$ |
| | | CONTRA | $0.950_{\pm.026}$ | $1.32_{\pm.719}$ | $0.953_{\pm.021}$ | $1.58_{\pm1.06}$ | $0.931_{\pm.036}$ | $6.21_{\pm4.51}$ |
| | | Local Ellipsoid | $0.970_{\pm.007}$ | $2.04_{\pm.505}$ | $0.975_{\pm.005}$ | $2.95_{\pm1.06}$ | $0.980_{\pm.003}$ | $3.82_{\pm1.13}$ |
| | | Empirical Copula | $0.973_{\pm.006}$ | $2.35_{\pm.446}$ | $0.972_{\pm.004}$ | $5.61_{\pm1.48}$ | $0.970_{\pm.005}$ | $40.4_{\pm6.04}$ |
| | | Gaussian Copula | $0.973_{\pm.006}$ | $2.37_{\pm.430}$ | $0.972_{\pm.004}$ | $5.61_{\pm1.48}$ | $0.970_{\pm.005}$ | $40.4_{\pm6.04}$ |
| | | TFT | $0.407_{\pm.065}$ | $0.292_{\pm.089}$ | $0.189_{\pm.306}$ | $0.07_{\pm.031}$ | $0.09_{\pm.007}$ | $0.009_{\pm.007}$ |
| | | DeepAR | $0.443_{\pm.095}$ | $0.308_{\pm.088}$ | $0.197_{\pm.054}$ | $0.07_{\pm.050}$ | $0.09_{\pm.028}$ | $0.004_{\pm.003}$ |
| | LSTM | FCP | $0.968_{\pm.022}$ | $\mathbf{0.859}_{\pm.075}$ | $0.966_{\pm.022}$ | $\mathbf{1.05}_{\pm.111}$ | $0.950_{\pm.010}$ | $\mathbf{1.82}_{\pm.287}$ |
| | | MultiDimSPCI | $0.957_{\pm.007}$ | $0.870_{\pm.383}$ | $0.960_{\pm.009}$ | $1.59_{\pm.588}$ | $0.952_{\pm.014}$ | $14.2_{\pm7.56}$ |
| | | CopulaCPTS | $1.000_{\pm.000}$ | $21.9_{\pm12.7}$ | $1.000_{\pm.000}$ | $330.0_{\pm219.4}$ | $0.992_{\pm.002}$ | $4.47\times10^4_{\pm4.23\times10^4}$ |
| | | OT-CP | $0.953_{\pm.006}$ | $0.920_{\pm.379}$ | $0.939_{\pm.027}$ | $11.8_{\pm9.35}$ | $0.921_{\pm.029}$ | $730.2_{\pm698.7}$ |
| | | CONTRA | $0.940_{\pm.258}$ | $0.222_{\pm.082}$ | $0.942_{\pm.028}$ | $0.106_{\pm.056}$ | $0.910_{\pm.032}$ | $0.050_{\pm.050}$ |
| | | Local Ellipsoid | $0.957_{\pm.023}$ | $0.987_{\pm.413}$ | $0.948_{\pm.008}$ | $1.48_{\pm.559}$ | $0.928_{\pm.017}$ | $3.37_{\pm.605}$ |
| | | Empirical Copula | $0.955_{\pm.005}$ | $3.81_{\pm.629}$ | $0.948_{\pm.010}$ | $25.8_{\pm5.06}$ | $0.920_{\pm.017}$ | $1.22\times10^3_{\pm281.9}$ |
| | | Gaussian Copula | $0.953_{\pm.006}$ | $3.74_{\pm.570}$ | $0.952_{\pm.011}$ | $26.4_{\pm4.00}$ | $0.920_{\pm.017}$ | $1.22\times10^3_{\pm281.9}$ |
| | | TFT | $0.374_{\pm.110}$ | $0.285_{\pm.106}$ | $0.192_{\pm.048}$ | $0.06_{\pm.022}$ | $0.062_{\pm.015}$ | $0.003_{\pm.002}$ |
| | | DeepAR | $0.386_{\pm.065}$ | $0.266_{\pm.069}$ | $0.211_{\pm.056}$ | $0.06_{\pm.017}$ | $0.09_{\pm.009}$ | $0.003_{\pm.001}$ |
| **Solar** | LOO Bootstrap | FCP | $0.957_{\pm.007}$ | $\mathbf{1.48}_{\pm.292}$ | $0.969_{\pm.003}$ | $\mathbf{4.18}_{\pm.597}$ | - | - |
| | | MultiDimSPCI | $0.968_{\pm.005}$ | $1.97_{\pm.076}$ | $0.971_{\pm.003}$ | $11.4_{\pm1.20}$ | - | - |
| | | CopulaCPTS | $1.000_{\pm.000}$ | $67.9_{\pm12.6}$ | $1.000_{\pm.000}$ | $7.25\times10^3_{\pm1.86\times10^3}$ | - | - |
| | | OT-CP | $0.984_{\pm.004}$ | $3.69_{\pm.797}$ | $0.971_{\pm.006}$ | $248.9_{\pm40.3}$ | - | - |
| | | CONTRA | $0.950_{\pm.012}$ | $3.08_{\pm.584}$ | $0.936_{\pm.013}$ | $30.8_{\pm16.7}$ | - | - |
| | | Local Ellipsoid | $0.947_{\pm.004}$ | $1.44_{\pm.188}$ | $0.948_{\pm.005}$ | $1.87_{\pm.540}$ | - | - |
| | | Empirical Copula | $0.986_{\pm.004}$ | $4.47_{\pm.174}$ | $0.988_{\pm.004}$ | $36.5_{\pm4.03}$ | - | - |
| | | Gaussian Copula | $0.986_{\pm.004}$ | $4.47_{\pm.174}$ | $0.989_{\pm.003}$ | $38.2_{\pm1.37}$ | - | - |
| | | TFT | $0.782_{\pm.026}$ | $0.779_{\pm.056}$ | $0.722_{\pm.028}$ | $3.18_{\pm.415}$ | - | - |
| | | DeepAR | $0.802_{\pm.121}$ | $1.03_{\pm.114}$ | $0.713_{\pm.086}$ | $6.73_{\pm1.09}$ | - | - |
| | LSTM | FCP | $0.968_{\pm.009}$ | $\mathbf{1.16}_{\pm.092}$ | $0.961_{\pm.008}$ | $\mathbf{2.09}_{\pm.566}$ | - | - |
| | | MultiDimSPCI | $0.969_{\pm.004}$ | $1.31_{\pm.010}$ | $0.976_{\pm.005}$ | $6.46_{\pm2.51}$ | - | - |
| | | CopulaCPTS | $1.000_{\pm.000}$ | $44.8_{\pm9.88}$ | $1.000_{\pm.000}$ | $3.34\times10^3_{\pm570}$ | - | - |
| | | OT-CP | $0.979_{\pm.005}$ | $2.25_{\pm.247}$ | $0.963_{\pm.008}$ | $142.0_{\pm40.8}$ | - | - |
| | | CONTRA | $0.938_{\pm.012}$ | $0.100_{\pm.026}$ | $0.913_{\pm.013}$ | $0.022_{\pm.014}$ | - | - |
| | | Local Ellipsoid | $0.972_{\pm.005}$ | $1.27_{\pm.143}$ | $0.978_{\pm.004}$ | $2.43_{\pm.996}$ | - | - |
| | | Empirical Copula | $0.987_{\pm.002}$ | $6.47_{\pm.103}$ | $0.990_{\pm.003}$ | $67.7_{\pm10.9}$ | - | - |
| | | Gaussian Copula | $0.992_{\pm.001}$ | $7.11_{\pm.216}$ | $0.997_{\pm.001}$ | $89.9_{\pm4.69}$ | - | - |
| | | TFT | $0.746_{\pm.081}$ | $0.651_{\pm.095}$ | $0.684_{\pm.063}$ | $1.63_{\pm.177}$ | - | - |
| | | DeepAR | $0.839_{\pm.028}$ | $1.01_{\pm.088}$ | $0.715_{\pm.043}$ | $3.57_{\pm.493}$ | - | - |

**Results.** Table 1 presents the results of experiments on three real-world datasets. FCP consistently obtains smaller prediction sets than all baselines while maintaining the target coverage. The performance gains of FCP are especially notable for higher outcome dimensions, showing significantly smaller prediction set sizes with lower variability. Moreover, FCP maintains stable coverage across varying $d_y$, whereas baseline methods often suffer from undercoverage or overcoverage coupled with either overly contracted or inflated prediction sets. In particular, methods relying on the exchangeability assumption often exhibit severe miscoverages and unstable prediction set sizes.

MultiDimSPCI and CP using local ellipsoids generally show good performance. In particular, on the solar dataset, CP with local ellipsoids achieves performance comparable to FCP. This is possibly due to their ability to capture temporal or local correlations, respectively. OT-CP and CONTRA also perform well in certain experiments, indicating some potential to adapt beyond the exchangeability assumption. We observe that increasing the guidance scale $w$ often reduces the prediction set size, though at the cost of slightly lower coverage. In practice, an effective range for $w$ is typically between 1 and 1.5 across our experiments.

**Ablation study.** We conduct an ablation study to assess the impact of the encoder. Specifically, we evaluate FCP with and without the encoder, where in the latter case the guidance is replaced by the concatenation of the feature at time $i$ and residual at time $i - 1$. Table 2 reports the average empirical coverage and prediction set sizes of FCP with and without the encoder on the wind dataset. We observe that removing the encoder led to less stable coverage and noticeably larger prediction set sizes.

Since the conditional coverage bound of FCP relies on the bi-Lipschitz flow assumption (Assumption 4.6), we conduct an additional ablation study using iResNet (Behrmann et al., 2019) to model the guided vector field, ensuring this assumption is satisfied. Table 7 reports the average empirical coverage and prediction set sizes of FCP with MLP and iResNet across the three datasets with varying $d_y$. We observe that imposing bi-Lipschitzness in the vector field does not degrade coverage or prediction set size.

Table 2: Average empirical coverage and prediction sets size obtained by FCP with and without the encoder on the wind dataset, evaluated under different base predictors and varying outcome dimensions $d_y$. Reported values represent the average and standard deviation over five independent experiments conducted on five constructed sequences. The target confidence level is set to 0.95.

| Base Predictor | Method | $d_y = 2$ | | $d_y = 4$ | | $d_y = 8$ | |
|---|---|---|---|---|---|---|---|
| | | Coverage | Size | Coverage | Size | Coverage | Size |
| LOO Bootstrap | FCP with Encoder | $0.951_{\pm.018}$ | $0.88_{\pm.089}$ | $0.953_{\pm.006}$ | $3.43_{\pm1.37}$ | $0.956_{\pm.010}$ | $19.4_{\pm10.2}$ |
| | FCP w/o Encoder | $0.948_{\pm.023}$ | $1.13_{\pm.193}$ | $0.964_{\pm.005}$ | $3.99_{\pm1.03}$ | $0.964_{\pm.010}$ | $35.3_{\pm14.0}$ |
| LSTM | FCP with Encoder | $0.952_{\pm.054}$ | $1.18_{\pm.215}$ | $0.957_{\pm.022}$ | $10.8_{\pm1.05}$ | $0.953_{\pm.056}$ | $2.48 \times 10^3_{\pm669}$ |
| | FCP w/o Encoder | $0.965_{\pm.011}$ | $1.92_{\pm.367}$ | $0.957_{\pm.014}$ | $12.2_{\pm15.0}$ | $0.935_{\pm.007}$ | $5.55 \times 10^3_{\pm7.47\times10^3}$ |

# 6 CONCLUSION

In this study, we proposed a novel conformal prediction method for multi-dimensional time series using flow with classifier-free guidance. We provided coverage guarantees of our method by establishing exact non-asymptotic marginal coverage and a finite-sample bound on conditional coverage. Experiments on real-world datasets with a broad set of baselines demonstrated that our method constructs smaller prediction sets while satisfying the target coverage, consistently outperforming the baselines. Future work will focus on investigating flow with optimal transport coupling to construct more efficient prediction sets, and on conducting extensive evaluations across diverse datasets with higher-dimensional outcomes.

## ACKNOWLEDGMENTS

The authors would like to thank Hanyang Jiang and Jonghyeok Lee for the insightful discussions. This work is partially supported by NSF CAREER CCF-1650913, NSF DMS-2134037, CMMI-2015787, CMMI-2112533, DMS-1938106, DMS-1830210, and the Coca-Cola Foundation.

## ETHICS STATEMENT

We confirm that our study complies with the ICLR Code of Ethics and does not present additional ethical issues.

## REPRODUCIBILITY STATEMENT

The source code for the method and experiments is available at https://github.com/Jayaos/flow_cp. Details of the experimental setup, including hyperparameters, datasets, are provided in the Experiments section and the Appendix B.

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

## A  PROOFS

**Proposition A.1.** *Let $u_{t|x_1}(x \mid x_1)$ be the vector field generating the probability path $p_{t|x_1}(x \mid x_1)$. Then, the vector field $u_{t|h}(x \mid h)$ is a valid vector field generating $p_{t|h}(x \mid h)$.*

*Proof.* Since $u_{t|x_1}(x \mid x_1)$ generates the probability path $p_{t|x_1}(x \mid x_1)$, the continuity equation holds as follows:

$$\frac{\partial}{\partial t} p_{t|x_1}(x \mid x_1) + \text{div}\left(u_{t|x_1}(x \mid x_1) p_{t|x_1}(x \mid x_1)\right) = 0. \tag{26}$$

By the definition of the guided marginal probability path, we have:

$$p_{t|h}(x \mid h) = \int p_{t|x_1}(x \mid x_1) q(x_1 \mid h) dx_1. \tag{27}$$

The time derivative of $p_{t|h}(x \mid h)$ is expressed as:

$$
\begin{aligned}
\frac{\partial}{\partial t} p_{t|h}(x \mid h) &= \frac{\partial}{\partial t} \int p_{t|x_1}(x \mid x_1) q(x_1 \mid h) \, dx_1 \\
&= \int \frac{\partial}{\partial t} p_{t|x_1}(x \mid x_1) q(x_1 \mid h) \, dx_1 \\
&= -\int \text{div}\left(u_{t|x_1}(x \mid x_1) p_{t|x_1}(x \mid x_1)\right) q(x_1 \mid h) \, dx_1 \\
&= -\text{div} \int u_{t|x_1}(x \mid x_1) p_{t|x_1}(x \mid x_1) q(x_1 \mid h) \, dx_1.
\end{aligned}
\tag{28}
$$

The marginal guided vector field is defined as follows:

$$u_{t|h}(x \mid h) := \int u_{t|x_1}(x \mid x_1) \frac{p_{t|x_1}(x \mid x_1) q(x_1 \mid h)}{p_{t|h}(x \mid h)} \, dx_1. \tag{29}$$

We can rewrite the marginal vector field as:

$$u_{t|h}(x \mid h) p_{t|h}(x \mid h) = \int u_{t|x_1}(x \mid x_1) p_{t|x_1}(x \mid x_1) q(x_1 \mid h) \, dx_1. \tag{30}$$

Substituting equation (30) into equation (28), we have:

$$\frac{\partial}{\partial t} p_{t|h}(x \mid h) = -\text{div}\left(u_{t|h}(x \mid h) p_{t|h}(x \mid h)\right), \tag{31}$$

which is the continuity equation for $p_{t|h}(x \mid h)$ under the guided vector field $u_{t|h}(x \mid h)$. Therefore, $u_{t|h}(x \mid h)$ is a valid vector field generating $p_{t|h}(x \mid h)$.  □

**Proposition A.2.** *With a given Gaussian probability path $p_{t|x_1}(x \mid x_1) = \mathcal{N}(x \mid \alpha_t x_1, \sigma_t^2 I_d)$, the guided vector field $u_{t|h}(x \mid h)$ can be reformulated as:*

$$u_{t|h}(x \mid h) = u_t(x) + b_t \nabla_x \log p_{h|t}(h \mid x). \tag{32}$$

*Proof.* By the definition of the guided marginal probability path, we have:

$$p_{t|h}(x \mid h) = \int p_{t|x_1}(x \mid x_1) q(x_1 \mid h) dx_1, \tag{33}$$

where $p_{t|x_1}(x \mid x_1) = \mathcal{N}(x \mid \alpha_t x_1, \sigma_t^2 I)$.

We can express the score function of the guided marginal probability path as:

$$\nabla_x \log p_{t|h}(x \mid h) = \frac{\nabla_x p_{t|h}(x \mid h)}{p_{t|h}(x \mid h)} \tag{34}$$

$$\overset{(i)}{=} \int \frac{\nabla_x p_{t|x_1}(x \mid x_1) q(x_1 \mid h) dx_1}{p_{t|h}(x \mid h)} \tag{35}$$

$$= \int \nabla_x \log p_{t|x_1}(x \mid x_1) \frac{p_{t|x_1}(x \mid x_1) q(x_1 \mid h)}{p_{t|h}(x \mid h)} dx_1, \tag{36}$$

where $(i)$ follows from the definition of the guided marginal probability path.

Since $p_{t|x_1}(x \mid x_1) = \mathcal{N}(x \mid \alpha_t x_1, \sigma_t^2 I)$, we have:

$$u_{t|x_1}(x \mid x_1) = \frac{\dot{\alpha}_t}{\sigma_t}(x - \alpha_t x_1) + \dot{\alpha}_t x_1 \tag{37}$$

$$= \frac{\dot{\alpha}_t}{\sigma_t} x - \frac{\dot{\alpha}_t}{\sigma_t} \alpha_t x_1 + \dot{\alpha}_t x_1 \tag{38}$$

$$= \frac{\dot{\alpha}_t}{\sigma_t} x + (\dot{\alpha}_t - \frac{\dot{\alpha}_t}{\sigma_t} \alpha_t) x_1 \tag{39}$$

$$= \frac{\dot{\alpha}_t}{\alpha_t} x + (\dot{\alpha}_t \sigma_t - \alpha_t \dot{\sigma}_t) \frac{1}{\alpha_t \sigma_t}(x - \alpha_t x_1) \tag{40}$$

$$= \frac{\dot{\alpha}_t}{\alpha_t} x + (\dot{\alpha}_t \sigma_t - \alpha_t \dot{\sigma}_t) \frac{\sigma_t}{\alpha_t} \nabla_x \log p_{t|x_1}(x \mid x_1), \tag{41}$$

where $\dot{\alpha}_t$ denotes $\frac{d}{dt} \alpha_t$, and $\dot{\sigma}_t$ denotes $\frac{d}{dt} \sigma_t$. The last equality holds since $\nabla_x \log p_{t|x_1}(x \mid x_1) = -\frac{1}{\sigma_t^2}(x - \alpha_t x_1)$.

The guided vector field is defined as:

$$u_{t|h}(x \mid h) = \int u_{t|x_1}(x \mid x_1) \frac{p_{t|x_1}(x \mid x_1) q(x_1 \mid h)}{p_{t|h}(x \mid h)} dx_1. \tag{42}$$

Therefore, by pugging $u_{t|x_1}(x \mid x_1)$ into equation (42), we have:

$$u_{t|h}(x \mid h) = a_t x + b_t \nabla_x \log p_t(x \mid h), \tag{43}$$

where $a_t = \frac{\dot{\alpha}_t}{\alpha_t}$, and $b_t = (\dot{\alpha}_t \sigma_t - \alpha_t \dot{\sigma}_t) \frac{\sigma_t}{\alpha_t}$.

By using the identity $\nabla_x \log p_{t|h}(x \mid h) = \nabla_x \log p_{h|t}(h \mid x) + \nabla_x \log p_t(x)$, we obtain:

$$u_t(x \mid h) = a_t x + b_t \left( \nabla \log p_{h|t}(h \mid x) + \nabla \log p_t(x) \right) = u_t(x) + b_t \nabla_x \log p_{h|t}(h \mid x). \tag{44}$$

$\square$

**Proposition A.3.** *The log-determinant Jacobian ODE defined in equation (16) is equivalent to the divergence of the guided vector field.*

*Proof.* The Jacobian ODE is defined as:

$$\frac{d}{dt} J_{\psi_{t|h}}(x \mid h) = \frac{\partial u_{t|h}(\psi_{t|h}(x \mid h))}{\partial \psi_{t|h}(x \mid h)} \frac{\partial \psi_{t|h}(x \mid h)}{\partial x} = \frac{\partial u_{t|h}(\psi_{t|h}(x \mid h))}{\partial \psi_{t|h}(x \mid h)} J_{\psi_{t|h}}(x \mid h), \quad (45)$$

with the initial condition:

$$J_{\psi_{t=0|h}}(x \mid h) = I. \quad (46)$$

By using Jacobi's formula, we have:

$$\frac{d}{dt} \det J_{\psi_{t|h}}(x \mid h) = \det J_{\psi_{t|h}}(x \mid h) \cdot \operatorname{tr}\left( J_{\psi_{t|h}}^{-1}(x \mid h) \frac{d}{dt} J_{\psi_{t|h}}(x \mid h) \right). \quad (47)$$

Substituting equation (45) into equation (47), we obtain:

$$\frac{d}{dt} \det J_{\psi_{t|h}}(x \mid h) = \det J_{\psi_{t|h}}(x \mid h) \cdot \operatorname{tr}\left( \frac{\partial u_{t|h}(\psi_{t|h}(x \mid h))}{\partial \psi_{t|h}(x \mid h)} \right). \quad (48)$$

Therefore,

$$\frac{d}{dt} \log |\det J_{\psi_{t|h}}(x \mid h)| = \operatorname{tr}\left( \frac{\partial u_{t|h}(\psi_{t|h}(x \mid h))}{\partial \psi_{t|h}(x \mid h)} \right). \quad (49)$$

Since the trace of the Jacobian of a vector field corresponds to its divergence, we have:

$$\operatorname{tr}\left( \frac{\partial u_{t|h}(\psi_{t|h}(x \mid h))}{\partial \psi_{t|h}(x \mid h)} \right) = \operatorname{div}\left( u_{t|h}(\psi_{t|h}(x \mid h)) \right). \quad (50)$$

Therefore, the log-determinant of the Jacobian ODE is defined as:

$$\frac{d}{dt} \log |\det J_{\psi_{t|h}}(x \mid h)| = \operatorname{div}\left( u_{t|h}(\psi_{t|h}(x \mid h)) \right) \quad (51)$$

with the initial condition:

$$\log |\det J_{\psi_{t=0|h}}(x \mid h)| = 0. \quad (52)$$

$$\square$$

**Theorem A.4** (Continuous images of connected sets, Rudin (1953)). *Let $X$ and $Y$ be topological spaces and $\psi : X \to Y$ be continuous. If $E \subset X$ is connected, then $\psi(E)$ is connected.*

**Theorem A.5** (Closed sets under homeomorphism, Munkres (2000)). *Let $X$ and $Y$ be topological spaces and $\psi : X \to Y$ be a homeomorphism. If $E \subset X$ is closed, then $\psi(E) \subset Y$ is closed.*

**Lemma A.6** (Lipschitz continuous of the guided flow). *Let $\psi_t$ denote the guided flow defined by a guided vector field $u_t$. If the guided vector field $u_t(x \mid h)$ is Lipschitz continuous in $x$ for all $t$ and $h$, i.e., there exists a constant $L_u > 0$ such that*

$$\|u_t(x \mid h) - u_t(x' \mid h)\| \le L_u \|x - x'\| \quad \forall x, x', t, h, \quad (53)$$

*then the guided flow $\psi_t(x \mid h)$ is Lipschitz continuous in $x$ for all $t$ and $h$. That is, there exists a constant $L_\psi > 0$ such that*

$$\|\psi_t(x \mid h) - \psi_t(x' \mid h)\| \le L_\psi \|x - x'\| \quad \forall x, x', t, h. \quad (54)$$

*Proof.* Let $d(t) = \|\psi_t(x \mid h) - \psi_t(x' \mid h)\|$ and $z(t) = \psi_t(x \mid h) - \psi_t(x' \mid h)$. Since the guided vector field is Lipschitz continuous, there exists $L_u$ such that

$$\|u_t(x \mid h) - u_t(x' \mid h)\| \le L_u \|x - x'\|, \quad \forall t, h, x, x'. \quad (55)$$

This is equivalent to

$$\|u_t(\psi_t(x \mid h) \mid h) - u_t(\psi_t(x' \mid h) \mid h)\| \le L_u \|\psi_t(x \mid h) - \psi_t(x' \mid h)\|, \quad \forall t, h, x, x'. \quad (56)$$

Since $d(t) \neq 0$, we have:

$$\frac{d}{dt}d(t) = \frac{1}{\|z(t)\|}\langle z(t), \frac{d}{dt}z(t)\rangle = \langle \frac{z(t)}{\|z(t)\|}, \frac{d}{dt}z(t)\rangle. \tag{57}$$

Since $\frac{d}{dt}z(t) = u_t(\psi_t(x \mid h) \mid h) - u_t(\psi_t(x' \mid h) \mid h)$, by Cauchy-Schwarz inequality, we have:

$$|\langle \frac{z(t)}{\|z(t)\|}, \frac{d}{dt}z(t)\rangle| \leq \|u_t(\psi_t(x \mid h) \mid h) - u_t(\psi_t(x' \mid h) \mid h)\|, \tag{58}$$

which implies:

$$\frac{d}{dt}d(t) \leq \|u_t(\psi_t(x \mid h) \mid h) - u_t(\psi_t(x' \mid h) \mid h)\| \leq L_u d(t), \tag{59}$$

where the last inequality follows from the Lipschitz continuity of the guided vector field.

Based on Gronwall's inequality (Gronwall, 1919; Hirsch et al., 2013), it follows that

$$d(t) \leq d(0)\, e^{L_u t}. \tag{60}$$

This can be proved by defining $\phi(t) := e^{-L_u t}d(t)$. Then,

$$\frac{d}{dt}\phi(t) = e^{-L_u t}\left(\frac{d}{dt}d(t) - L_u d(t)\right) \leq 0, \tag{61}$$

which implies $\phi$ is non-increasing on $t$. Since $\phi(t) \leq \phi(0) = d(0)$ for all $t \in [0, 1]$, multiplying both sides by $e^{L_u t}$ yields:

$$d(t) \leq d(0)e^{L_u t}. \tag{62}$$

Therefore,

$$\log d(t) - \log d(0) \leq L_u t. \tag{63}$$

Exponentiating both sides gives

$$\frac{d(t)}{d(0)} \leq e^{L_u t}, \tag{64}$$

and hence

$$d(t) \leq d(0)e^{L_u t}. \tag{65}$$

Since $d(0) = \|\psi_0(x \mid h) - \psi_0(x' \mid h)\| = \|x - x'\|$, we conclude that

$$\|\psi_t(x \mid h) - \psi_t(x' \mid h)\| \leq e^{L_u t}\|x - x'\|. \tag{66}$$

$\square$

***Proof of Lemma 4.3***. Since the probability density function of $Y = \psi(X)$ is the push-forward of $p_X$, we have:

$$p_Y(y) = p_X(\psi^{-1}(y)) \left|\det J_{\psi^{-1}}(y)\right|, \tag{67}$$

where $\det A$ denotes the determinant $A$ and $J_{\psi^{-1}}(y) = \frac{\partial \psi^{-1}(y)}{\partial y}$ is the Jacobian of $\psi^{-1}$. The probability mass of the transformed set $\mathcal{A}' = \psi(\mathcal{A})$ is:

$$\mathbb{P}(Y \in \mathcal{A}') = \int_{\mathcal{A}'} p_Y(y)\, dy. \tag{68}$$

Using the change-of-variables $y = \psi(x)$ with $dy = |\det J_\psi(x)|dx$, we have:

$$\int_{\mathcal{A}'} p_Y(y)\, dy = \int_{\mathcal{A}} p_Y(\psi(x))\left|\det J_\psi(x)\right|\, dx. \tag{69}$$

Substituting from equation (67), we have:

$$\int_{\mathcal{A}} p_Y(\psi(x))\left|\det J_\psi(x)\right|\, dx = \int_{\mathcal{A}} p_X(x)\left|\det J_{\psi^{-1}}(\psi(x))\right|\left|\det J_\psi(x)\right|\, dx. \tag{70}$$

Since $J_{\psi^{-1}}(\psi(x)) = J_\psi(x)^{-1}$, we know that $|\det J_{\psi^{-1}}(\psi(x))| \cdot |\det J_\psi(x)| = 1$. Hence,

$$\int_{\mathcal{A}'} p_Y(y)\, dy = \int_{\mathcal{A}} p_X(x)\, dx. \tag{71}$$

$\square$

**Lemma A.7** (bi-Lipschitz guided flow). *Assume that the guided vector field is bi-Lipschitz in $x$ over for all $t$ and $h$, i.e., there exists $L_u$ and $l_u$ such that*

$$l_u\|x - x'\| \leq \|u_t(x \mid h) - u_t(x' \mid h)\| \leq L_u\|x - x'\| \quad \forall t, h, x, x'. \tag{72}$$

*Then the guided flow $\psi$ is bi-Lipschitz. There exists $L_\psi$ and $l_\psi$ such that*

$$l_\psi\|x - x'\| \leq \|\psi_t(x \mid h) - \psi_t(x' \mid h)\| \leq L_\psi\|x - x'\| \quad \forall t, h, x, x'. \tag{73}$$

*Proof.* Proof follows similarly to Lemma A.6.

Let $z(t) = \psi_t(x \mid h) - \psi_t(x' \mid h)$ and $d(t) = \|\psi_t(x \mid h) - \psi_t(x' \mid h)\| = \|z_t\|$.

$$\frac{d}{dt}\|z(t)\|^2 = 2\langle z(t), \frac{d}{dt}z(t)\rangle \tag{74}$$

By Cauchy-Schwarz inequality,

$$\frac{d}{dt}\|z(t)\|^2 = \frac{d}{dt}d(t)^2 \geq -2\|z(t)\|\|\frac{d}{dt}z(t)\| \tag{75}$$

Since $\frac{d}{dt}z(t) = u_t(x \mid h) - u_t(x' \mid h)$ and $\|u_t(x \mid h) - u_t(x' \mid h)\| \geq l_u\|x - x'\| = l_u\|\psi_t(x \mid h) - \psi_t(x' \mid h)\|$, we have:

$$\frac{d}{dt}d(t)^2 \geq -2l_u\|z(t)\|^2 = -2l_u d(t)^2 \tag{76}$$

Using Gronwall's inequality,

$$\|\psi_t(x \mid h) - \psi_t(x' \mid h)\| \geq e^{-l_u t}\|x - x'\| \tag{77}$$

Therefore, we know that

$$\|\psi_t(x \mid h) - \psi_t(x' \mid h)\| \geq e^{-l_u}\|x - x'\| \quad \forall x, x', t, h \tag{78}$$

Combining with the upper Lipschitz bound obtained from Lemma A.6, we conclude that

$$e^{-l_u}\|x - x'\| \leq \|\psi_t(x \mid h) - \psi_t(x' \mid h)\| \leq e^{L_u}\|x - x'\| \quad \forall x, x', t, h \tag{79}$$

$\square$

**Lemma A.8.** *Under Assumption 4.8, $F_e(e_{T+1}) \sim \mathrm{Unif}[0, 1]$.*

*Proof.* Since $F_e$ is strictly increasing and continuous under Assumption 4.8, the Lemma holds for $e_{T+1} \sim F_e$. $\square$

**Lemma A.9** (Convergence of empirical CDF of i.i.d. $\{e_i\}_{i=1}^T$). *Under Assumption 4.5 and 4.6, for any $T$, there exists an event $A_T$ with probability at least $1 - \sqrt{\frac{\log(16T)}{T}}$, such that conditioned on $A_T$,*

$$\sup_x \left|\tilde{F}_{T+1}(x) - F_e(x)\right| \leq \sqrt{\frac{\log(16T)}{T}}. \tag{80}$$

*Proof.* This proof follows the proof of Lemma 1 in Xu & Xie (2023a). Under the assumption that $\{e_i\}_{i=1}^{T+1}$ are i.i.d., the Dvoretzky–Kiefer–Wolfowitz (DKW) inequality (Dvoretzky et al., 1956; Kosorok, 2008) implies:

$$\mathbb{P}\left(\sup_x \left|\tilde{F}_{T+1}(x) - F_e(x)\right| > s_T\right) \leq 2e^{-2Ts_T^2}. \tag{81}$$

Choose $s_T = \sqrt{W(16T)/(2\sqrt{T})}$, where $W(T)$ denotes the Lambert $W$ function satisfying $W(T)e^{W(T)} = T$. Since $W(16T) \leq \log(16T)$, it follows that $s_T \leq \sqrt{\log(16T)/T}$. Define the event $A_T$ on which $\sup_x \left| \tilde{F}_{T+1}(x) - F_e(x) \right| \leq \sqrt{\log(16T)/T}$, so that we have:

$$\sup_x \left| \tilde{F}_{T+1}(x) - F_e(x) \right| \Big| A_T \leq \sqrt{\frac{\log(16T)}{T}}, \tag{82}$$

and

$$\mathbb{P}(A_T) > 1 - \sqrt{\frac{\log(16T)}{T}}. \tag{83}$$

$\square$

**Lemma A.10** (Gaussian concentration inequality, Theorem 5.6 in Boucheron et al. (2003)). *Let $X \sim \mathcal{N}(0, I_d)$ be a standard Gaussian random vector in $\mathbb{R}^d$ and let $f : \mathbb{R}^d \to \mathbb{R}$ be an $L_f$-Lipschitz continuous function. Then, for all $t > 0$,*

$$\mathbb{P}(f(X) \geq \mathbb{E}[f(X)] + t) \leq \exp\left(\frac{-t^2}{2L_f{}^2}\right). \tag{84}$$

**Proposition A.11** (Gaussian concentration inequality with isotropic covariance). *Let $X \sim \mathcal{N}(0, \gamma I_d)$ be an isotropic Gaussian random vector in $\mathbb{R}^d$ with covariance matrix $\gamma I_d \in \mathbb{R}^d$ for some $\gamma > 0$. Let $f : \mathbb{R}^d \to \mathbb{R}$ be an $L_f$-Lipschitz continuous function. Then, for all $t > 0$,*

$$\mathbb{P}(f(X) \geq \mathbb{E}[f(X)] + t) \leq \exp\left(\frac{-t^2}{2\gamma L_f{}^2}\right). \tag{85}$$

*Proof.* Let $X' \sim \mathcal{N}(0, I_d)$, and define $X = \sqrt{\gamma} X'$, so that $X \sim \mathcal{N}(0, \gamma I_d)$. Define the function $f_\gamma(x) := f(\sqrt{\gamma} x)$. Then $f_\gamma$ is $\sqrt{\gamma} L_f$-Lipschitz. Using Lemma A.10 to $f_\gamma(X')$, we obtain:

$$\mathbb{P}\left(f_\gamma(X') \geq \mathbb{E}[f_\gamma(X')] + t\right) \leq \exp\left(\frac{-t^2}{2\gamma L_f{}^2}\right). \tag{86}$$

Since $f(X) = f_\gamma(X')$,

$$\mathbb{P}(f(X) \geq \mathbb{E}[f(X)] + t) = \mathbb{P}\left(f_\gamma(X') \geq \mathbb{E}[f_\gamma(X')] + t\right) \leq \exp\left(\frac{-t^2}{2\gamma L_f{}^2}\right). \tag{87}$$

$\square$

**Lemma A.12** (Norm concentration of isotropic Gaussian random vectors). *Let $X_i \sim \mathcal{N}(\boldsymbol{0}, \gamma I_d)$ be an isotropic Gaussian random vector in $\mathbb{R}^d$, and $\| \cdot \|$ be 2-norm. Then for any $\delta \in (0, 1)$, with probability at least $1 - \delta$, we have:*

$$\max_{1 \leq i \leq T} \|X_i\| \leq M_T, \tag{88}$$

*where $M_T = \sqrt{\gamma} \left( \sqrt{d} + \sqrt{2\log(T/\delta)} \right)$.*

*Proof.* Using Proposition A.11 and since $f$ is 1-Lipschitz continuous, we have for all $t > 0$:

$$\mathbb{P}(\|X\| \geq \mathbb{E}[\|X\|] + t) \leq \exp\left(-\frac{t^2}{2\gamma}\right). \tag{89}$$

Using Jensen's inequality and since $X \sim \mathcal{N}(0, \gamma I_d)$,

$$\mathbb{E}[\|X\|] \leq \sqrt{\mathbb{E}[\|X\|^2]} = \sqrt{\mathbb{E}[X^\top X]} = \sqrt{\operatorname{tr}(\gamma I_d)} = \sqrt{\gamma d}. \tag{90}$$

Therefore, for any $t > 0$,

$$\mathbb{P}\left(\|X\| \geq \sqrt{\gamma d} + t\right) \leq \exp\left(-\frac{t^2}{2\gamma}\right). \tag{91}$$

By the union bound,

$$\mathbb{P}\left(\max_{1 \leq i \leq T} \|X_i\| \geq \sqrt{\gamma d} + t\right) \leq \sum_{i=1}^{T} \mathbb{P}\left(\|X_i\| \geq \sqrt{\gamma d} + t\right) \leq T \cdot \exp\left(-\frac{t^2}{2\gamma}\right). \tag{92}$$

By setting $T \cdot \exp\left(-t^2/2\gamma\right) \leq \delta$, we obtain:

$$t \geq \sqrt{2\gamma \log\left(\frac{T}{\delta}\right)}. \tag{93}$$

Therefore, with probability at least $1 - \delta$,

$$\max_{1 \leq i \leq T} \|X_i\| \leq \sqrt{\gamma d} + \sqrt{2\gamma \log\left(\frac{T}{\delta}\right)}. \tag{94}$$

Defining $M_T := \sqrt{\gamma}\left(\sqrt{d} + \sqrt{2\log(T/\delta)}\right)$, we conclude:

$$\max_{1 \leq i \leq T} \|X_i\| \leq M_T. \tag{95}$$

$\square$

**Lemma A.13** (Bound on the sum of differences between true and estimated non-conformity scores)**.**
*Under Assumption 4.6 and 4.9, with probability at least $1 - \delta$,*

$$\sum_{i=1}^{T} |\hat{e}_i - e_i| \leq T(2M_T L_{\psi^{-1}}\delta_T + L_{\psi^{-1}}^2 \delta_T^2). \tag{96}$$

*Proof.* Since the encoder is fixed after convergence, it generates the same $h$ for $\hat{\epsilon}$ and $\epsilon$. Let $\hat{s}_i = \psi^{-1}(\hat{\epsilon}_i \mid h)$ and $s_i = \psi^{-1}(\epsilon_i \mid h)$.

Using the identity for the difference of squared norms, we have:

$$\begin{aligned}
\|\hat{s}_i\| &= \|s_i + (\hat{s}_i - s_i)\|^2 \\
&= \|s_i\|^2 + 2\langle s_i, \hat{s}_i - s_i\rangle + \|\hat{s}_i - s_i\|^2.
\end{aligned} \tag{97}$$

Therefore,

$$\|\hat{s}_i\|^2 - \|s_i\|^2 = 2\langle s_i, \hat{s}_i - s_i\rangle + \|\hat{s}_i - s_i\|^2, \tag{98}$$

which implies

$$\begin{aligned}
|\hat{e}_i - e_i| &= \left|\|\hat{s}_i\|^2 - \|s_i\|^2\right| \\
&= \left|2\langle s_i, \hat{s}_i - s_i\rangle + \|\hat{s}_i - s_i\|^2\right|.
\end{aligned} \tag{99}$$

By the Cauchy-Schwarz inequality,

$$|\langle s_i, \hat{s}_i - s_i\rangle| \leq \|s_i\| \cdot \|\hat{s}_i - s_i\|. \tag{100}$$

Since $\psi^{-1}$ is Lipschitz continuous with Lipschitz constant $L_{\psi^{-1}}$, we have:

$$\|\hat{s}_i - s_i\| \leq L_{\psi^{-1}}\|\hat{\epsilon}_i - \epsilon_i\| = L_{\psi^{-1}}\|\Delta_i\|. \tag{101}$$

Substituting inequality (101) into the inner product bound in equation (100),

$$|\langle s_i, \hat{s}_i - s_i \rangle| \le \|s_i\| \cdot \|\hat{s}_i - s_i\| \le L_{\psi^{-1}} \|s_i\| \|\Delta_i\|. \tag{102}$$

Then, by the triangle inequality,

$$|\hat{e}_i - e_i| \le 2L_{\psi^{-1}} \|s_i\| \|\Delta_i\| + L_{\psi^{-1}}^2 \|\Delta_i\|^2. \tag{103}$$

By Lemma A.12, we have with probability at least $1 - \delta$ that $\|s_i\| \le M_T$ for all $i$, and by Assumption 4.9, $\|\Delta_i\| \le \delta_T$. Substituting these into the inequality (103), we have:

$$|\hat{e}_i - e_i| \le 2M_T L_{\psi^{-1}} \delta_T + L_{\psi^{-1}}^2 \delta_T^2. \tag{104}$$

Summing over all $i = 1, \ldots, T$, we conclude that

$$\sum_{i=1}^{T} |\hat{e}_i - e_i| \le T\left(2M_T L_{\psi^{-1}} \delta_T + L_{\psi^{-1}}^2 \delta_T^2\right). \tag{105}$$

$\square$

**Lemma A.14** (Distance between the empirical CDF of $\{e_i\}_{i=1}^T$ and $\{\hat{e}_i\}_{i=1}^T$). *Under Assumption 4.6, 4.8, and 4.9, with probability $1 - \delta$, $\hat{F}_{T+1}(x)$ and $\tilde{F}_{T+1}(x)$ satisfy*

$$\sup_x \left|\hat{F}_{T+1}(x) - \tilde{F}_{T+1}(x)\right| \le (2L_{T+1} + 1)C + 2\sup_x \left|\tilde{F}_{T+1}(x) - F_e(x)\right|, \tag{106}$$

*where $C = \sqrt{M_T L_{\psi^{-1}} \delta_T + L_{\psi^{-1}}^2 \delta_T^2}$.*

*Proof.* By Lemma A.13, with probability at least $1 - \delta$, we have:

$$\sum_{t=1}^{T} |\hat{e}_t - e_t| \le T\left(2M_T L_{\psi^{-1}} \delta_T + L_{\psi^{-1}}^2 \delta_T^2\right). \tag{107}$$

Let $C = \left(2M_T L_{\psi^{-1}} \delta_T + L_{\psi^{-1}}^2 \delta_T^2\right)^{1/2}$. Then,

$$\sum_{i=1}^{T} |\hat{e}_i - e_i| \le TC^2. \tag{108}$$

Define $S = \{t : |\hat{e}_t - e_t| \ge C\}$. Then,

$$|S| \cdot C \le \sum_{t=1}^{T} |\hat{e}_t - e_t| \le TC^2, \tag{109}$$

which implies $|S| \le TC$.

We can bound the difference between the empirical CDFs of $\hat{e}_i$ and $e_i$ as follows:

$$
\begin{aligned}
|\widehat{F}_{T+1}(x) - \widetilde{F}_{T+1}(x)| &\leq \frac{1}{T}\sum_{t=1}^{T} |\mathbb{1}\{\hat{e}_t \leq x\} - \mathbb{1}\{e_t \leq x\}| \\
&\leq \frac{1}{T}\left(|S| + \sum_{t \notin S} |\mathbb{1}\{\hat{e}_t \leq x\} - \mathbb{1}\{e_t \leq x\}|\right) \\
&\overset{(i)}{\leq} \frac{1}{T}\left(|S| + \sum_{t \notin S} \mathbb{1}\{|e_t - x| \leq C\}\right) \\
&\leq \frac{1}{T}\left(|S| + \sum_{t=1}^{T} \mathbb{1}\{|e_t - x| \leq C\}\right) \\
&\leq C + \mathbb{P}(|e_{T+1} - x| \leq C) \\
&\quad + \sup_{x}\left|\frac{1}{T}\sum_{t=1}^{T} \mathbb{1}\{|e_t - x| \leq C\} - \mathbb{P}(|e_{T+1} - x| \leq C)\right| \\
&\overset{(ii)}{=} C + [F_e(x+C) - F_e(x-C)] \\
&\quad + \sup_{x}\left|\left[\widetilde{F}_{T+1}(x+C) - \widetilde{F}_{T+1}(x-C)\right] - [F_e(x+C) - F_e(x-C)]\right| \\
&\overset{(iii)}{\leq} (2L_{T+1}+1)C + 2\sup_{x}|\widetilde{F}_{T+1}(x) - F_e(x)|.
\end{aligned}
$$
(110)

Here, $(i)$ follows from the inequality $|\mathbb{1}\{a \leq x\} - \mathbb{1}\{b \leq x\}| \leq \mathbb{1}\{|b-x| \leq |a-b|\}$ for $a, b \in \mathbb{R}$, $(ii)$ follows from the identity $\mathbb{P}(|e_{T+1} - x| \leq C) = F_e(x+C) - F_e(x-C)$, and $(iii)$ uses the Lipschitz continuity of $F_e(x)$.

$\square$

***Proof of Theorem 4.10.*** For any $\beta \in [0, \alpha]$,

$$
\begin{aligned}
&\left|\mathbb{P}\left(Y_{T+1} \in \widehat{C}_{T+1}^{\alpha} \mid Z_{T+1} = z_{T+1}\right) - (1-\alpha)\right| \\
&= \left|\mathbb{P}\left(\hat{e}_{T+1} \in \left[\widehat{F}_{T+1}^{-1}(\beta), \widehat{F}_{T+1}^{-1}(1-\alpha+\beta)\right] \mid Z_{T+1} = z_{T+1}\right) - (1-\alpha)\right| \\
&\overset{(i)}{=} \left|\mathbb{P}\left(\beta \leq \widehat{F}_{T+1}(\hat{e}_{T+1}) \leq 1-\alpha+\beta\right) - \mathbb{P}\left(\beta \leq F_e(e_{T+1}) \leq 1-\alpha+\beta\right)\right|.
\end{aligned}
$$
(111)

Equality $(i)$ follows from Lemma A.8, which states that $F_e(e_{T+1}) \sim \text{Unif}[0,1]$. This can be further bounded by:

$$
\begin{aligned}
&\left|\mathbb{P}\left(\beta \leq \widehat{F}_{T+1}(\hat{e}_{T+1}) \leq 1-\alpha+\beta\right) - \mathbb{P}\left(\beta \leq F_e(e_{T+1}) \leq 1-\alpha+\beta\right)\right| \\
&\leq \mathbb{E}\left|\mathbb{1}\left\{\beta \leq \widehat{F}_{T+1}(\hat{e}_{T+1}) \leq 1-\alpha+\beta\right\} - \mathbb{1}\left\{\beta \leq F_e(e_{T+1}) \leq 1-\alpha+\beta\right\}\right| \\
&\overset{(i)}{\leq} \mathbb{E}\left(\left|\mathbb{1}\left\{\beta \leq \widehat{F}_{T+1}(\hat{e}_{T+1})\right\} - \mathbb{1}\left\{\beta \leq F_e(e_{T+1})\right\}\right| \right. \\
&\quad \left. + \left|\mathbb{1}\left\{\widehat{F}_{T+1}(\hat{e}_{T+1}) \leq 1-\alpha+\beta\right\} - \mathbb{1}\left\{F_e(e_{T+1}) \leq 1-\alpha+\beta\right\}\right|\right)
\end{aligned}
$$
(112)

Here, inequality $(i)$ follows from the fact that for any $a, b \in \mathbb{R}$ and real values $x, y \in \mathbb{R}$,

$$
|\mathbb{1}\{a \leq x \leq b\} - \mathbb{1}\{a \leq y \leq b\}| \leq |\mathbb{1}\{a \leq x\} - \mathbb{1}\{a \leq y\}| + |\mathbb{1}\{x \leq b\} - \mathbb{1}\{y \leq b\}|. \quad (113)
$$

By triangle inequality,

$$\mathbb{E}\left(\left|\mathbb{1}\left\{\beta \leq \widehat{F}_{T+1}(\hat{e}_{T+1})\right\} - \mathbb{1}\left\{\beta \leq F_e(e_{T+1})\right\}\right|\right.$$
$$+ \left.\left|\mathbb{1}\left\{\widehat{F}_{T+1}(\hat{e}_{T+1}) \leq 1 - \alpha + \beta\right\} - \mathbb{1}\left\{F_e(e_{T+1}) \leq 1 - \alpha + \beta\right\}\right|\right)$$
$$\leq \underbrace{\mathbb{E}\left(\left|\mathbb{1}\{\beta \leq \widehat{F}_{T+1}(\hat{e}_{T+1})\} - \mathbb{1}\{\beta \leq F_e(e_{T+1})\}\right|\right)}_{(a)} \tag{114}$$
$$+ \underbrace{\mathbb{E}\left(\left|\mathbb{1}\left\{\widehat{F}_{T+1}(\hat{e}_{T+1}) \leq 1 - \alpha + \beta\right\} - \mathbb{1}\left\{F_e(e_{T+1}) \leq 1 - \alpha + \beta\right\}\right|\right)}_{(b)}$$

For term $(a)$, we have:

$$\mathbb{E}\left(\left|\mathbb{1}\{\beta \leq \widehat{F}_{T+1}(\hat{e}_{T+1})\} - \mathbb{1}\{\beta \leq F_e(e_{T+1})\}\right|\right)$$
$$\leq \mathbb{P}\left(|F_e(e_{T+1}) - \beta| \leq |\widehat{F}_{T+1}(\hat{e}_{T+1}) - F_e(e_{T+1})|\right). \tag{115}$$

This inequality follows from the fact that for $a, b \in \mathbb{R}$, $|\mathbb{1}\{a \leq x\} - \mathbb{1}\{b \leq x\}| \leq \mathbb{1}\{|b - x| \leq |a - b|\}$, and $\mathbb{E}[\mathbb{1}\{A\}] = \mathbb{P}(A)$.

Similarly, for term (b), we have:

$$\mathbb{E}\left(\left|\mathbb{1}\left\{\widehat{F}_{T+1}(\hat{e}_{T+1}) \leq 1 - \alpha + \beta\right\} - \mathbb{1}\left\{F_e(e_{T+1}) \leq 1 - \alpha + \beta\right\}\right|\right)$$
$$\leq \mathbb{P}\left(|F_e(e_{T+1}) - (1 - \alpha + \beta)| \leq \left|\widehat{F}_{T+1}(\hat{e}_{T+1}) - F_e(e_{T+1})\right|\right). \tag{116}$$

Therefore,

$$\left|\mathbb{P}\left(Y_{T+1} \in \widehat{C}_{T+1}^{\alpha} \mid Z_{T+1} = z_{T+1}\right) - (1 - \alpha)\right|$$
$$\leq \mathbb{P}\left(|F_e(e_{T+1}) - \beta| \leq |\widehat{F}_{T+1}(\hat{e}_{T+1}) - F_e(e_{T+1})|\right) \tag{117}$$
$$+ \mathbb{P}\left(|F_e(e_{T+1}) - (1 - \alpha + \beta)| \leq |\widehat{F}_{T+1}(\hat{e}_{T+1}) - F_e(e_{T+1})|\right)$$

In Lemma A.9, we defined $A_T$ as the event on which

$$\sup_x |\tilde{F}_{T+1}(x) - F_e(x)|\big| A_T \leq \sqrt{\frac{\log(16T)}{T}},$$

where $\mathbb{P}(A_T) > 1 - \sqrt{\frac{\log(16T)}{T}}$. Let $A_T^C$ denote the complement of the event $A_T$. For any $\gamma \in [0, 1]$, we have:

$$\mathbb{P}\left(|F_e(e_{T+1}) - \gamma| \leq |\hat{F}_{T+1}(\hat{e}_{T+1}) - F_e(e_{T+1})|\right)$$
$$\leq \mathbb{P}\left(|F_e(e_{T+1}) - \gamma| \leq |\hat{F}_{T+1}(\hat{e}_{T+1}) - F_e(e_{T+1})| \mid A_T\right) + \mathbb{P}(A_T^C)$$
$$\leq \mathbb{P}\left(|F_e(e_{T+1}) - \gamma| \leq |\hat{F}_{T+1}(\hat{e}_{T+1}) - F_e(\hat{e}_{T+1})| + |F_e(\hat{e}_{T+1}) - F_e(e_{T+1})| \mid A_T\right) \tag{118}$$
$$+ \sqrt{\frac{\log(16T)}{T}}.$$

To bound the conditional probability above, we note that with probability $1 - \delta$, conditioning on the event $A_T$,

$$|\widehat{F}_{T+1}(\hat{e}_{T+1}) - F_e(e_{T+1})| + |F_e(\hat{e}_{T+1}) - F_e(e_{T+1})| \,\big|\, A_T$$

$$\overset{(i)}{\leq} \sup_x |\widehat{F}_{T+1}(x) - F_e(x)| \,\big|\, A_T + L_{T+1}|\hat{e}_{T+1} - e_{T+1}|$$

$$\leq \sup_x |\widehat{F}_{T+1}(x) - \widetilde{F}_{T+1}(x)| \,\big|\, A_T + \sup_x |\widetilde{F}_{T+1}(x) - F_e(x)| \,\big|\, A_T + L_{T+1}|\hat{e}_{T+1} - e_{T+1}| \tag{119}$$

$$\overset{(ii)}{\leq} (2L_{T+1} + 1)C + 3\sup_x |\widetilde{F}_{T+1}(x) - F_e(x)| \,\big|\, A_T + L_{T+1}\delta_T$$

$$\overset{(iii)}{\leq} 3\sqrt{\frac{\log(16T)}{T}} + \left(L_{T+1} + \frac{1}{2}\right)(2C + \delta_T).$$

Here, inequality $(i)$ holds due to the supremum of $|\widehat{F}_{T+1}(x) - F_e(x)|$ over $x$ and Lipschitz continuity of $F_e$ from Assumption 4.8. Inequality $(ii)$ follows from Lemma A.14. Inequality $(iii)$ follows from Lemma A.9.

Since $F_e(e_{T+1}) \sim \text{Unif}[0,1]$,

$$\mathbb{P}\left(|F_e(e_{T+1}) - \gamma| \leq \left|\widehat{F}_{T+1}(\hat{e}_{T+1}) - F_e(\hat{e}_{T+1})\right| + |F_e(\hat{e}_{T+1}) - F_e(e_{T+1})| \,\Big|\, A_T\right)$$

$$\leq 6\sqrt{\frac{\log(16T)}{T}} + 2\left(L_{T+1} + \frac{1}{2}\right)(2C + \delta_T). \tag{120}$$

Therefore, by substituting inequality (120) to inequality (117), we obtain:

$$\left|\mathbb{P}\left(Y_{T+1} \in \widehat{C}_{T+1}^\alpha \mid Z_{T+1} = z_{T+1}\right) - (1-\alpha)\right|$$

$$\leq 12\sqrt{\frac{\log(16T)}{T}} + 4(L_{T+1} + \frac{1}{2})(2C + \delta_T). \tag{121}$$

$\square$

**Definition A.15.** A sequence of random variables $\{X_n\}$ is said to be *strictly stationary* if for every $k \geq 1$, any integers $n_1, \ldots, n_k$, and any integer $h$, the joint distribution of the random variables $(X_{n_1}, \ldots, X_{n_k})$ is the same as the joint distribution of $(X_{n_1+h}, \ldots, X_{n_k+h})$.

**Definition A.16.** A sequence of random variables $\{X_n\}$ is said to be *strongly mixing* (or $\alpha$-*mixing*) if the mixing coefficients $\alpha(k)$ defined by

$$\alpha(k) = \sup_{n \in \mathbb{N}} \sup_{A \in \mathcal{F}_1^n, B \in \mathcal{F}_{n+k}^\infty} |\mathbb{P}(A \cap B) - \mathbb{P}(A)\mathbb{P}(B)| \tag{122}$$

satisfy $\alpha(k) \to 0$ as $k \to \infty$, where $\mathcal{F}_a^b$ denotes the $\sigma$-algebra generated by $\{X_a, \ldots, X_b\}$.

**Lemma A.17** (Convergence of empirical CDF of stationary and strongly mixing $\{e_i\}_{i=1}^T$). *Under Assumption 4.11, for any $T$, there exists an event $A_T$ with probability at least $1 - (\frac{M(\log T)^2}{2T})^{1/3}$, such that conditioned on $A_T$,*

$$\sup_x |\tilde{F}_{T+1}(x) - F_e(x)| \leq \frac{(\frac{M}{2})^{1/3}(\log T)^{2/3}}{T^{1/3}}. \tag{123}$$

*Proof.* The proof follows similarly in the proof of Lemma B.11 in Xu et al. (2024). Define $v_T(x) := \sqrt{T}(\tilde{F}_{T+1}(x) - F_e(x))$. By using Proposition 7.1 in Rio et al. (2017), we have:

$$\mathbb{E}\left(\sup_x |v_T(x)|^2\right) \leq \left(1 + 4\sum_{k=0}^T \alpha(k)\right)\left(3 + \frac{\log T}{2\log 2}\right)^2, \tag{124}$$

where $\alpha(k)$ denotes the $k$-th mixing coefficient. Under Assumption 4.11, we have $\sum_{k \geq 0} \alpha(k) \leq M < \infty$. Applying Markov's inequality yields:

$$\mathbb{P}\left(\sup_x \left|\widetilde{F}_{T+1}(x) - F_e(x)\right| \geq s_T\right) \leq \frac{\mathbb{E}\left(\sup_x |v_T(x)|^2/T\right)}{s_T^2} \leq \frac{1+4M}{Ts_T^2}\left(3 + \frac{\log T}{2\log 2}\right)^2. \quad (125)$$

By setting

$$s_T := \left(\frac{1+4M}{T}\left(3 + \frac{\log T}{2\log 2}\right)^2\right)^{1/3} \approx \left(\frac{M(\log T)^2}{2T}\right)^{1/3}, \quad (126)$$

we then have:

$$\mathbb{P}\left(\sup_x \left|\widetilde{F}_{T+1}(x) - F_e(x)\right| \leq \left(\frac{M(\log T)^2}{2T}\right)^{1/3}\right) \geq 1 - \left(\frac{M(\log T)^2}{2T}\right)^{1/3}. \quad (127)$$

Define the event $A_T$ on which $\sup_x \left|\tilde{F}_{T+1}(x) - F_e(x)\right| \leq \left(\frac{M(\log T)^2}{2T}\right)^{1/3}$, so that we have:

$$\sup_x \left|\tilde{F}_{T+1}(x) - F_e(x)\right| \Big| A_T \leq \left(\frac{M(\log T)^2}{2T}\right)^{1/3} \quad (128)$$

and

$$\mathbb{P}(A_T) > 1 - \left(\frac{M(\log T)^2}{2T}\right)^{1/3}. \quad (129)$$

$\square$

***Proof of Corollary 4.12.*** Under Assumption 4.11, the result follows by combining Lemma A.14 and A.17, using an argument analogous to the proof of Theorem 4.10.

$\square$

## B  EXPERIMENT DETAILS

### B.1  EXPERIMENT SETUP

**OT-CP.** We implement OT-CP using the source code released by the authors (Thurin et al., 2025). The training and validation sets are combined to form a calibration set. Following the setup in the original publication, 75% of the calibration set is used to solve OT, and the remaining 25% is used to calibrate the prediction sets.

**CONTRA.** As the source code from the original publication (Fang et al., 2025) was not released, we implement CONTRA following the methodology and details provided in the original publication. Consistent with the original setup, we use six coupling layers with a hidden dimension of 128 and train the model for 100 epochs with the same batch size as FCP and a learning rate of 0.001. The training and validation sets are combined into a calibration set, of which 50% is used to train the model and the remaining 50% is used to calibrate the prediction sets.

**MultiDimSPCI.** We implement MultiDimSPCI (Xu et al., 2024) using the source code released by the authors. The context window size is set to 50 for all real-world datasets, consistent with the setup used for FCP. Following the original publication, the number of trees for the quantile random forest is set to 15. The training and validation sets are combined into a single training set.

**Conformal prediction using copulas.** We implement the method using the source code released by the authors (Messoudi et al., 2021). Following the setup described in the original publication, the training and validation sets are combined to form a calibration set.

**Conformal prediction using local ellipsoids**  We implement the method using the source code provided by the authors (Messoudi et al., 2022). Following the setup in the original publication, the training set is used as the proper training set and the validation set as the calibration set. The number of neighbors for kNN is set to 5% of the proper training set size, as suggested in the original publication. We also conduct experiments with different numbers of neighbors, but does not lead to meaningful differences in performance.

**CopulaCPTS**  We implemented CopulaCPTS using the source code provided by the authors (Sun & Yu, 2022). Following the setup described in the original publication, the training and validation sets are combined to form a calibration set.

**Temporal Fusion Transformer**  We implement Temporal Fusion Transformer (TFT) (Lim et al., 2021) using Pytorch Forecasting. A hyperparameter grid search is conducted on the training set of each dataset with $d_y = 2$ to determine the optimal configuration. We believe this hyperparameter search generalizes well to higher $d_y$ within each dataset, since TFT makes predictions for each outcome dimension independently in our setup. Performance is observed to saturate at a model dimension of 32, with two attention heads and two layers, therefore this setting is used for all experiments. For consistency with FCP, the context window size is fixed at 50 across all experiments. We train the models using the Adam (Kingma, 2014) with a learning rate of 0.001, a maximum of 50 epochs, and a dropout rate of 0.1. Quantile loss with $q \in \{0.025, 0.975\}$ is used for 0.95 target coverage.

**DeepAR**  We implement DeepAR (Salinas et al., 2020) using Pytorch Forecasting. A hyperparameter grid search is conducted on the training set of each dataset with $d_y = 2$ to determine the optimal configuration similarly to TFT. Performance is observed to saturate at a model dimension of 32 with two layers, therefore this setting is used for all experiments. For consistency with FCP, the context window size is fixed at 50 across all experiments. We train the models using the Adam (Kingma, 2014) with a learning rate of 0.001, a maximum of 50 epochs, and a dropout rate of 0.1. Multivariate normal distribution loss with $q \in \{0.025, 0.975\}$ is used for 0.95 target coverage.

Table 3: The hyperparameter search space for FCP.

|  | Hyperparameter | Search space |
|---|---|---|
| **Vector field** | the number of layers
hidden dimension | { 2, 4, 6 }
{ 16, 32, 64 } |
| **Encoder** | the number of layers
the number of heads
model dimension
dropout | { 2, 4, 6 }
{ 2, 4, 8 }
{ 16, 32, 64 }
{ 0, 0.1 } |
| **General** | covariance scale $\gamma$
learning rate
batch size | { 1, 2, 4, 8 }
{ 0.0005, 0.0001 }
{ 8, 16 } |

**FCP**  We use multilayer perceptions (MLP) to model the guided vector field. The time variable $t \in [0, 1]$ and guidance $h \in \mathbb{R}^{d_h}$ are concatenated with the input and fed into the guided vector field. A hyperparameter grid search is conducted on the training set of each dataset with different $d_y$ to determine the optimal configuration. We set the hidden dimension of the vector field identical to the model dimension of the encoder, so that additional layer is not required between the vector field and the encoder. Table 3 shows the hyperparameter search space and Table 4 shows the optimized hyperparameter configuration. The context window size for the encoder is set to 50. We train the model with Adam (Kingma, 2014) with a maximum of 50 epochs for all experiments and use the validation set to select the best model.

To determine an appropriate sample size $N$ for the set size estimation using quasi-Monte Carlo sampling, we compute the relative standard error of the Jacobian determinants of $\psi$, defined as $\text{SE}(\det J_{\psi,h})/\text{Avg}(\det J_{\psi,h})$, where $\det J_{\psi,h} = \{\det J_\psi(x_j \mid h)\}_{j=1}^N$ are the sampled Jacobian

determinants conditioned on $h$. We select the smallest $N$ such that the average relative standard error across all $h$ falls below 0.01. As a result, we use $N = 4096$ for experiments with $d_y = 2$, $N = 8192$ for experiments with $d_y = 4$, and $N = 16384$ for experiments with $d_y = 8$.

Table 4: The optimized hyperparameter configuration for FCP based on the hyperparameter search.

| Dataset | Hyperparameter | $\mathbf{d_y = 2}$ | $\mathbf{d_y = 4}$ | $\mathbf{d_y = 8}$ |
|---|---|---|---|---|
| Wind | the number of layers of the vector field | 4 | 4 | 4 |
| | the number of heads of the encoder | 2 | 2 | 2 |
| | the number of layers of the encoder | 4 | 4 | 4 |
| | the hidden dimension of the vector field and encoder | 32 | 32 | 32 |
| | covariance scale $\gamma$ | 1 | 1 | 2 |
| | encoder dropout | 0.1 | 0.1 | 0.1 |
| | batch size | 4 | 4 | 4 |
| | learning rate | 0.0005 | 0.0005 | 0.0005 |
| | null condition probability | 0.05 | 0.05 | 0.05 |
| | guidance scale $w$ (LOO/LSTM base predictor) | 1.1/1.1 | 1.1/1.1 | 1.1/1.1 |
| Traffic | the number of layers of the vector field | 4 | 4 | 4 |
| | the number of heads of the encoder | 2 | 2 | 2 |
| | the number of layers of the encoder | 4 | 4 | 4 |
| | the hidden dimension of the vector field and encoder | 32 | 32 | 32 |
| | covariance scale $\gamma$ | 1 | 1 | 1 |
| | encoder dropout | 0.1 | 0.1 | 0.1 |
| | batch size | 8 | 8 | 8 |
| | learning rate | 0.0001 | 0.0001 | 0.0001 |
| | null condition probability | 0.05 | 0.05 | 0.05 |
| | guidance scale $w$ (LOO/LSTM base predictor) | 1.1/1.2 | 1.1/1.2 | 1.05/1.5 |
| Solar | the number of layers of the vector field | 4 | 4 | - |
| | the number of heads of the encoder | 2 | 2 | - |
| | the number of layers of the encoder | 4 | 4 | - |
| | the hidden dimension of the vector field and encoder | 32 | 32 | - |
| | covariance scale $\gamma$ | 1 | 1 | - |
| | encoder dropout | 0.1 | 0.1 | - |
| | batch size | 8 | 8 | - |
| | learning rate | 0.0005 | 0.0005 | - |
| | null condition probability | 0.05 | 0.05 | - |
| | guidance scale $w$ (LOO/LSTM base predictor) | 1.5/1.2 | 1.2/1.1 | - |

## B.2 COMPUTATIONAL COST

**Training time.** Table 5 reports the wall-clock training time for all methods, computed as the sum over five independent runs. All models were trained on a machine equipped with dual Intel Xeon Gold 6226 CPUs and a single NVIDIA A100 GPU. For the methods that do not use neural networks, only CPU was used.

## C DATASET DETAILS

**Wind dataset** The wind dataset contains wind speed records measured at 30 different wind farms (Zhu et al., 2021). Each wind farm location provides 768 records with 5 features at each timestamp. We randomly select $d_y \in \{2, 4, 8\}$ locations to construct five sequences of $d_y$-dimensional time series.

**Traffic dataset** The traffic dataset contains traffic flow collected at 15 different traffic sensor locations (Xu & Xie, 2021b). Each sensor location provides 8778 observations with 5 features at each timestamp. We randomly select $d_y \in \{2, 4, 8\}$ locations to construct five sequences of $d_y$-dimensional time series.

Table 5: The wall-clock training time (hrs) for all methods, computed as the sum over five independent runs.

| Dataset | Method | $d_y=2$ | $d_y=4$ | $d_y=8$ |
|---|---|---|---|---|
| Wind | FCP | $\leq 0.2$ | $\leq 0.2$ | $\leq 0.2$ |
| | CONTRA | $\leq 0.2$ | $\leq 0.2$ | $\leq 0.2$ |
| | MultiDimSPCI | $\leq 0.05$ | $\leq 0.05$ | $\leq 0.05$ |
| | Local Ellipsoid | $\leq 0.01$ | $\leq 0.01$ | $\leq 0.01$ |
| | Empirical Copula | $\leq 0.01$ | $\leq 0.01$ | $\leq 0.01$ |
| | Gaussian Copula | $\leq 0.01$ | $\leq 0.01$ | $\leq 0.01$ |
| | CopulaCPTS | $\leq 0.01$ | $\leq 0.01$ | $\leq 0.01$ |
| | TFT | $\leq 1$ | $\leq 2$ | $\leq 4$ |
| | DeepAR | $\leq 1$ | $\leq 2$ | $\leq 4$ |
| Traffic | FCP | $\leq 0.5$ | $\leq 0.5$ | $\leq 0.5$ |
| | CONTRA | $\leq 0.5$ | $\leq 0.5$ | $\leq 0.5$ |
| | MultiDimSPCI | $\leq 1$ | $\leq 1$ | $\leq 1$ |
| | Local Ellipsoid | $\leq 0.01$ | $\leq 0.01$ | $\leq 0.01$ |
| | Empirical Copula | $\leq 0.01$ | $\leq 0.01$ | $\leq 0.01$ |
| | Gaussian Copula | $\leq 0.01$ | $\leq 0.01$ | $\leq 0.01$ |
| | CopulaCPTS | $\leq 0.01$ | $\leq 0.01$ | $\leq 0.01$ |
| | TFT | $\leq 4$ | $\leq 8$ | $\leq 16$ |
| | DeepAR | $\leq 4$ | $\leq 8$ | $\leq 16$ |
| Solar | FCP | $\leq 0.5$ | $\leq 0.5$ | – |
| | CONTRA | $\leq 0.5$ | $\leq 0.5$ | – |
| | MultiDimSPCI | $\leq 1$ | $\leq 1$ | – |
| | Local Ellipsoid | $\leq 0.01$ | $\leq 0.01$ | – |
| | Empirical Copula | $\leq 0.01$ | $\leq 0.01$ | – |
| | Gaussian Copula | $\leq 0.01$ | $\leq 0.01$ | – |
| | CopulaCPTS | $\leq 0.01$ | $\leq 0.01$ | – |
| | TFT | $\leq 4$ | $\leq 8$ | – |
| | DeepAR | $\leq 4$ | $\leq 8$ | – |

**Solar dataset** The solar dataset considers solar radiation in Diffused Horizontal Irradiance (DHI) units at 9 different solar sensor locations (Zhang et al., 2021). Each location provides 8755 records with 5 features at each timestamp. For the solar dataset, we randomly selected $d_y \in \{2, 4\}$ locations to construct five sequences of $d_y$-dimensional time series. We did not construct sequences with $d_y = 8$ due to the limited number of unique locations, which could lead to overlapping sequences across different trials of experiments.

# D  ADDITIONAL EXPERIMENTS

## D.1  EXPERIMENTS AT DIFFERENT CONFIDENCE LEVELS

Table 6 reports the results on the three real-world datasets at the 0.9 confidence level. We exclude TFT and DeepAR, as they did not demonstrate competitive performance in the experiment at the 0.95 confidence level. The overall results remain consistent with those at the 0.95 confidence level. Notably, the gap in average prediction set sizes between FCP and other strong baselines—such as MultiDimSPCI, CP using local ellipsoids, and OT-CP for $d_y \in 2, 4$ on the traffic and solar datasets—decreases at the 0.9 confidence level.

## D.2  ROLLING COVERAGE

Since conditional coverage is difficult to assess on real-world data, we use rolling coverage as a proxy for conditional coverage at a specific time index. The rolling coverage at time index $i$ is defined as:

$$\widehat{\mathrm{RC}}_i = \frac{1}{m} \sum_{j=0}^{m-1} \mathbb{1}\left\{ y_{i-j} \in \widehat{C}_{i-j}(z_{i-j}, \alpha) \right\}, \tag{130}$$

Table 6: Average empirical coverage and prediction sets size obtained by FCP and all baselines on three real-world datasets, evaluated under different base predictors and varying outcome dimensions $d_y$. Reported values represent the average and standard deviation over five independent experiments conducted on five constructed sequences. The target confidence level is set to 0.9.

| Dataset | Base Predictor | Method | $d_y = 2$ | | $d_y = 4$ | | $d_y = 8$ | |
|---|---|---|---|---|---|---|---|---|
| | | | Coverage | Size | Coverage | Size | Coverage | Size |
| **Wind** | LOO Bootstrap | FCP | $0.906_{\pm.022}$ | $0.596_{\pm.050}$ | $0.925_{\pm.017}$ | $0.734_{\pm.139}$ | $0.938_{\pm.011}$ | $5.24_{\pm1.45}$ |
| | | MultiDimSPCI | $0.917_{\pm.013}$ | $0.790_{\pm.341}$ | $0.919_{\pm.024}$ | $2.26_{\pm1.49}$ | $0.933_{\pm.015}$ | $47.7_{\pm52.5}$ |
| | | CopulaCPTS | $1.000_{\pm.000}$ | $22.3_{\pm19.0}$ | $1.000_{\pm.000}$ | $611.3_{\pm484.7}$ | $1.000_{\pm.000}$ | $3.50\times10^5_{\pm3.73\times10^5}$ |
| | | OT-CP | $0.919_{\pm.033}$ | $0.904_{\pm.572}$ | $0.951_{\pm.025}$ | $23.9_{\pm20.9}$ | $0.883_{\pm.025}$ | $1.00\times10^3_{\pm622.8}$ |
| | | CONTRA | $0.919_{\pm.045}$ | $6.53_{\pm5.17}$ | $0.974_{\pm.016}$ | $4.12\times10^4_{\pm5.05\times10^4}$ | $0.974_{\pm.016}$ | $4.12\times10^9_{\pm4.05\times10^9}$ |
| | | Local Ellipsoid | $0.943_{\pm.028}$ | $0.952_{\pm.409}$ | $0.958_{\pm.015}$ | $3.58_{\pm2.18}$ | $0.961_{\pm.008}$ | $53.2_{\pm68.1}$ |
| | | Empirical Copula | $0.914_{\pm.023}$ | $0.597_{\pm.204}$ | $0.917_{\pm.021}$ | $1.21_{\pm.375}$ | $0.896_{\pm.042}$ | $7.38_{\pm2.04}$ |
| | | Gaussian Copula | $0.914_{\pm.023}$ | $0.622_{\pm.189}$ | $0.917_{\pm.021}$ | $1.54_{\pm.725}$ | $0.919_{\pm.019}$ | $17.0_{\pm4.48}$ |
| | LSTM | FCP | $0.917_{\pm.061}$ | $0.884_{\pm.161}$ | $0.924_{\pm.024}$ | $5.72_{\pm.718}$ | $0.896_{\pm.065}$ | $848.4_{\pm229.2}$ |
| | | MultiDimSPCI | $0.948_{\pm.022}$ | $2.68_{\pm1.15}$ | $0.904_{\pm.040}$ | $41.9_{\pm46.8}$ | $0.839_{\pm.074}$ | $2.37\times10^3_{\pm2.16\times10^3}$ |
| | | CopulaCPTS | $1.000_{\pm.000}$ | $45.7_{\pm45.4}$ | $1.000_{\pm.000}$ | $4.82\times10^3_{\pm3.73\times10^3}$ | $1.000_{\pm.000}$ | $2.83\times10^7_{\pm3.28\times10^7}$ |
| | | OT-CP | $0.909_{\pm.046}$ | $5.98_{\pm2.84}$ | $0.900_{\pm.029}$ | $188.1_{\pm106.3}$ | $0.978_{\pm.019}$ | $7.21\times10^4_{\pm3.49\times10^4}$ |
| | | CONTRA | $0.730_{\pm.240}$ | $0.22_{\pm.202}$ | $0.696_{\pm.247}$ | $0.05_{\pm.023}$ | $0.761_{\pm.177}$ | $7.71_{\pm6.80}$ |
| | | Local Ellipsoid | $0.978_{\pm.043}$ | $7.40_{\pm4.25}$ | $1.000_{\pm.000}$ | $167.3_{\pm137.5}$ | $1.000_{\pm.000}$ | $1.28\times10^5_{\pm1.24\times10^5}$ |
| | | Empirical Copula | $0.974_{\pm.042}$ | $10.6_{\pm5.93}$ | $1.000_{\pm.000}$ | $325.9_{\pm148.9}$ | $0.991_{\pm.017}$ | $2.38\times10^5_{\pm5.90\times10^4}$ |
| | | Gaussian Copula | $0.978_{\pm.043}$ | $10.7_{\pm5.86}$ | $1.000_{\pm.000}$ | $331.4_{\pm131.8}$ | $0.991_{\pm.017}$ | $3.01\times10^5_{\pm1.17\times10^5}$ |
| **Traffic** | LOO Bootstrap | FCP | $0.913_{\pm.026}$ | $0.613_{\pm.243}$ | $0.935_{\pm.010}$ | $0.453_{\pm.223}$ | $0.934_{\pm.039}$ | $1.03_{\pm.101}$ |
| | | MultiDimSPCI | $0.920_{\pm.008}$ | $1.01_{\pm.262}$ | $0.929_{\pm.011}$ | $1.48_{\pm.468}$ | $0.934_{\pm.006}$ | $2.92_{\pm.911}$ |
| | | CopulaCPTS | $1.000_{\pm.000}$ | $21.6_{\pm16.3}$ | $1.000_{\pm.000}$ | $645.8_{\pm645.5}$ | $1.000_{\pm.000}$ | $3.18\times10^5_{\pm4.80\times10^5}$ |
| | | OT-CP | $0.921_{\pm.008}$ | $1.09_{\pm.269}$ | $0.927_{\pm.010}$ | $2.39_{\pm.915}$ | $0.914_{\pm.006}$ | $1.46\times10^3_{\pm588.0}$ |
| | | CONTRA | $0.892_{\pm.037}$ | $0.606_{\pm.325}$ | $0.902_{\pm.034}$ | $0.565_{\pm.317}$ | $0.849_{\pm.048}$ | $0.414_{\pm.309}$ |
| | | Local Ellipsoid | $0.927_{\pm.021}$ | $1.22_{\pm.391}$ | $0.942_{\pm.010}$ | $1.17_{\pm.391}$ | $0.945_{\pm.008}$ | $0.954_{\pm.376}$ |
| | | Empirical Copula | $0.915_{\pm.013}$ | $1.24_{\pm.296}$ | $0.930_{\pm.004}$ | $2.17_{\pm.399}$ | $0.931_{\pm.004}$ | $9.63_{\pm3.17}$ |
| | | Gaussian Copula | $0.915_{\pm.012}$ | $1.26_{\pm.294}$ | $0.934_{\pm.007}$ | $2.38_{\pm.501}$ | $0.936_{\pm.008}$ | $10.9_{\pm1.68}$ |
| | LSTM | FCP | $0.953_{\pm.022}$ | $0.633_{\pm.148}$ | $0.945_{\pm.019}$ | $0.623_{\pm.058}$ | $0.923_{\pm.032}$ | $0.673_{\pm.298}$ |
| | | MultiDimSPCI | $0.914_{\pm.008}$ | $0.607_{\pm.255}$ | $0.914_{\pm.014}$ | $0.977_{\pm.388}$ | $0.913_{\pm.022}$ | $4.82_{\pm2.70}$ |
| | | CopulaCPTS | $1.000_{\pm.000}$ | $21.9_{\pm12.7}$ | $1.000_{\pm.000}$ | $330.0_{\pm219.4}$ | $0.999_{\pm.002}$ | $4.47\times10^5_{\pm4.25\times10^5}$ |
| | | OT-CP | $0.894_{\pm.007}$ | $0.575_{\pm.238}$ | $0.875_{\pm.025}$ | $1.99_{\pm1.26}$ | $0.850_{\pm.042}$ | $356.5_{\pm322.9}$ |
| | | CONTRA | $0.889_{\pm.025}$ | $0.129_{\pm.050}$ | $0.860_{\pm.043}$ | $0.031_{\pm.020}$ | $0.809_{\pm.060}$ | $0.007_{\pm.006}$ |
| | | Local Ellipsoid | $0.915_{\pm.028}$ | $0.625_{\pm.262}$ | $0.899_{\pm.021}$ | $0.706_{\pm.325}$ | $0.871_{\pm.039}$ | $1.12_{\pm.341}$ |
| | | Empirical Copula | $0.908_{\pm.015}$ | $2.59_{\pm.383}$ | $0.912_{\pm.019}$ | $13.9_{\pm2.72}$ | $0.880_{\pm.020}$ | $515.2_{\pm105.7}$ |
| | | Gaussian Copula | $0.910_{\pm.017}$ | $2.62_{\pm.363}$ | $0.908_{\pm.017}$ | $13.3_{\pm2.69}$ | $0.874_{\pm.019}$ | $479.0_{\pm141.1}$ |
| **Solar** | LOO Bootstrap | FCP | $0.905_{\pm.014}$ | $0.589_{\pm.109}$ | $0.900_{\pm.010}$ | $1.67_{\pm.326}$ | - | - |
| | | MultiDimSPCI | $0.930_{\pm.007}$ | $1.10_{\pm.068}$ | $0.942_{\pm.006}$ | $5.13_{\pm.435}$ | - | - |
| | | CopulaCPTS | $1.000_{\pm.000}$ | $67.9_{\pm12.6}$ | $1.000_{\pm.000}$ | $7.25\times10^3_{\pm1.86\times10^3}$ | - | - |
| | | OT-CP | $0.936_{\pm.016}$ | $1.44_{\pm.440}$ | $0.928_{\pm.009}$ | $8.54_{\pm1.84}$ | - | - |
| | | CONTRA | $0.889_{\pm.004}$ | $1.38_{\pm.506}$ | $0.878_{\pm.010}$ | $7.16_{\pm4.09}$ | - | - |
| | | Local Ellipsoid | $0.897_{\pm.010}$ | $0.749_{\pm.064}$ | $0.885_{\pm.010}$ | $0.320_{\pm.059}$ | - | - |
| | | Empirical Copula | $0.949_{\pm.007}$ | $1.98_{\pm.192}$ | $0.955_{\pm.005}$ | $7.87_{\pm.909}$ | - | - |
| | | Gaussian Copula | $0.953_{\pm.005}$ | $2.12_{\pm.142}$ | $0.962_{\pm.004}$ | $9.66_{\pm.626}$ | - | - |
| | LSTM | FCP | $0.911_{\pm.051}$ | $0.673_{\pm.288}$ | $0.907_{\pm.016}$ | $0.535_{\pm.104}$ | - | - |
| | | MultiDimSPCI | $0.938_{\pm.006}$ | $0.733_{\pm.066}$ | $0.937_{\pm.004}$ | $2.60_{\pm1.04}$ | - | - |
| | | CopulaCPTS | $1.000_{\pm.000}$ | $44.8_{\pm9.88}$ | $1.000_{\pm.000}$ | $3.34\times10^3_{\pm570.7}$ | - | - |
| | | OT-CP | $0.914_{\pm.011}$ | $0.585_{\pm.084}$ | $0.924_{\pm.019}$ | $10.6_{\pm5.80}$ | - | - |
| | | CONTRA | $0.835_{\pm.021}$ | $0.112_{\pm.037}$ | $0.858_{\pm.017}$ | $0.034_{\pm.033}$ | - | - |
| | | Local Ellipsoid | $0.921_{\pm.012}$ | $0.582_{\pm.055}$ | $0.934_{\pm.005}$ | $0.514_{\pm.250}$ | - | - |
| | | Empirical Copula | $0.925_{\pm.005}$ | $2.98_{\pm.082}$ | $0.939_{\pm.010}$ | $17.3_{\pm4.44}$ | - | - |
| | | Gaussian Copula | $0.939_{\pm.002}$ | $3.56_{\pm.203}$ | $0.964_{\pm.005}$ | $28.0_{\pm2.64}$ | - | - |

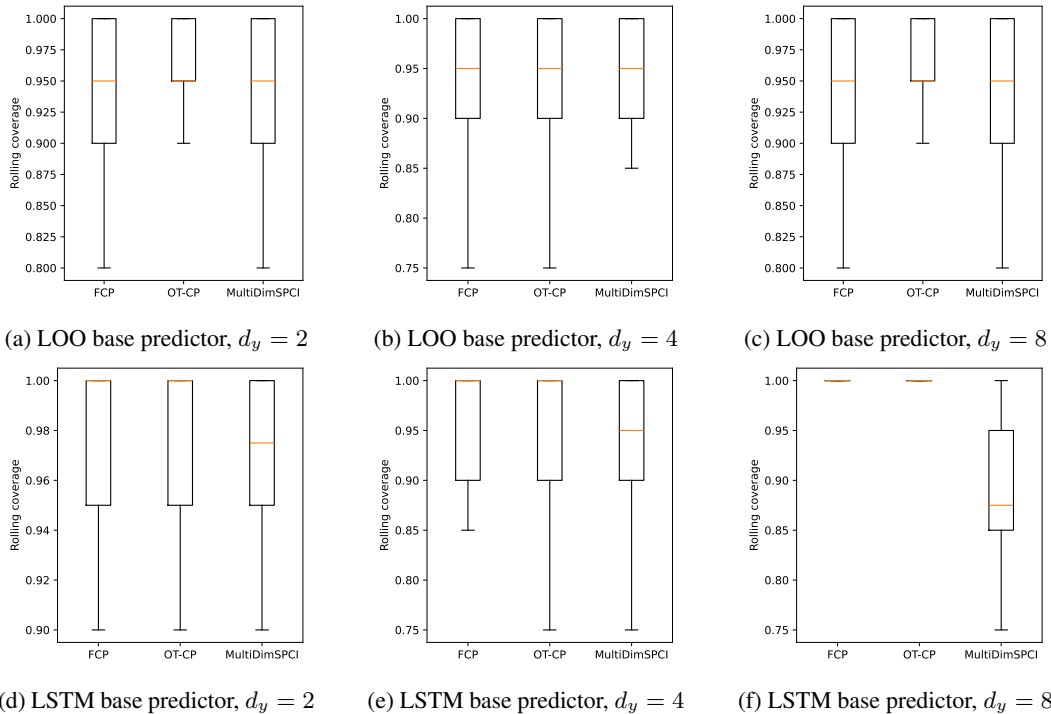

(a) LOO base predictor, $d_y = 2$      (b) LOO base predictor, $d_y = 4$      (c) LOO base predictor, $d_y = 8$

(d) LSTM base predictor, $d_y = 2$      (e) LSTM base predictor, $d_y = 4$      (f) LSTM base predictor, $d_y = 8$

Figure 3: Boxplots of the rolling coverages on the wind dataset with rolling window size 20 for FCP, OT-CP, and MultiDimSPCI.

Table 7: Average empirical coverage and prediction sets sizes obtained by FCP using MLP vector field and iResNet vector field on three real-world datasets, evaluated under different base predictors and varying outcome dimensions $d_y$. Reported values represent the average and standard deviation over five independent experiments. The target confidence level was set to 0.95. Results with average empirical coverage below the target confidence level are grayed out, and the smallest prediction set sizes, excluding the grayed-out results, are highlighted in bold.

| Dataset | Base Predictor | Method | $\mathbf{d_y = 2}$ | | $\mathbf{d_y = 4}$ | | $\mathbf{d_y = 8}$ | |
|---|---|---|---|---|---|---|---|---|
| | | | Coverage | Size | Coverage | Size | Coverage | Size |
| **Wind** | LOO Bootstrap | FCP (MLP) | $0.951_{\pm.018}$ | $\mathbf{0.88}_{\pm.089}$ | $0.953_{\pm.006}$ | $3.43_{\pm1.37}$ | $0.956_{\pm.010}$ | $19.4_{\pm10.2}$ |
| | | FCP (iResNet) | $0.951_{\pm.021}$ | $1.14_{\pm.069}$ | $0.954_{\pm.014}$ | $\mathbf{1.79}_{\pm.736}$ | $0.953_{\pm.018}$ | $\mathbf{14.8}_{\pm22.5}$ |
| | LSTM | FCP (MLP) | $0.952_{\pm.054}$ | $\mathbf{1.18}_{\pm.215}$ | $0.957_{\pm.022}$ | $10.8_{\pm1.05}$ | $0.953_{\pm.056}$ | $\mathbf{2.48 \times 10^3}_{\pm669}$ |
| | | FCP (iResNet) | $0.957_{\pm.034}$ | $1.84_{\pm.279}$ | $0.957_{\pm.018}$ | $\mathbf{6.37}_{\pm2.91}$ | $0.978_{\pm.015}$ | $2.55 \times 10^3_{\pm1.94\times10^3}$ |
| **Traffic** | LOO Bootstrap | FCP (MLP) | $0.957_{\pm.014}$ | $\mathbf{0.915}_{\pm.119}$ | $0.953_{\pm.009}$ | $\mathbf{1.06}_{\pm.431}$ | $0.965_{\pm.015}$ | $\mathbf{1.53}_{\pm.161}$ |
| | | FCP (iResNet) | $0.950_{\pm.021}$ | $1.21_{\pm.084}$ | $0.959_{\pm.014}$ | $1.33_{\pm.118}$ | $0.970_{\pm.007}$ | $2.72_{\pm.215}$ |
| | LSTM | FCP (MLP) | $0.968_{\pm.022}$ | $0.859_{\pm.075}$ | $0.966_{\pm.022}$ | $\mathbf{1.05}_{\pm.111}$ | $0.950_{\pm.010}$ | $\mathbf{1.82}_{\pm.287}$ |
| | | FCP (iResNet) | $0.957_{\pm.024}$ | $\mathbf{0.788}_{\pm.051}$ | $0.970_{\pm.010}$ | $1.31_{\pm.103}$ | $0.956_{\pm.016}$ | $2.50_{\pm.328}$ |
| **Solar** | LOO Bootstrap | FCP (MLP) | $0.957_{\pm.007}$ | $1.48_{\pm.292}$ | $0.969_{\pm.003}$ | $4.18_{\pm.597}$ | - | - |
| | | FCP (iResNet) | $0.952_{\pm.009}$ | $\mathbf{1.42}_{\pm.166}$ | $0.956_{\pm.003}$ | $\mathbf{2.69}_{\pm.196}$ | - | - |
| | LSTM | FCP (MLP) | $0.968_{\pm.009}$ | $\mathbf{1.16}_{\pm.092}$ | $0.961_{\pm.008}$ | $\mathbf{2.09}_{\pm.566}$ | - | - |
| | | FCP (iResNet) | $0.955_{\pm.005}$ | $1.24_{\pm.076}$ | $0.955_{\pm.008}$ | $2.42_{\pm.276}$ | - | - |

where $m$ is a rolling window size. Figure 3 presents boxplots of $\widehat{\mathrm{RC}}_i$ computed with $m = 20$ on the wind dataset.

