# OpenReview forum: "Flow-based Conformal Prediction for Multi-dimensional Time Series"
_ICLR.cc/2026/Conference — ICLR 2026 Poster_

### Official Review · Reviewer_W2dB · 2025-10-27

**Soundness:** 3
**Presentation:** 3
**Contribution:** 3
**Rating:** 6
**Confidence:** 4

**Summary:**

This paper introduces a conformal prediction method for multi-dimensional time series that addresses two main challenges: temporal dependencies and multi-output prediction. The approach combines a Flow model with Transformer-based context encoding to model residual distributions, using Euclidean distance for non-conformity scoring. Trained via Flow Matching, the method provides theoretical coverage guarantees and achieves tighter prediction sets than baselines on real-world datasets.

**Strengths:**

1. The integration of flow matching with conformal prediction is original and conceptually appealing.
2. The method provides solid coverage guarantees, both marginal and conditional.
3. Experimental results show significant improvements over baselines.
4. The paper is well written and easy to follow, with clear motivation and presentation.

**Weaknesses:**

1. While flow matching may help model the distribution of prediction residuals, it can be computationally expensive.
2. Results may be sensitive to hyperparameters, and extensive hyperparameter search can further increase computational cost.

**Questions:**

1. Please compare computational cost against baselines and report sensitivity to hyperparameters.
2. Please empirically validate Assumptions 4.1, 4.3, and 4.7–4.11 in the experiments (or provide diagnostics/proxies).
3. I'm not sure about the key reason that the flow-based model yields substantially smaller prediction sets than baselines. What properties of flows drive the gains, and why alternatives like quantile regression or transformer encoders cannot achieve similar improvements?

---

> ### Author Response · Authors · 2025-11-25
> **Rebuttal (1/2)**
>
> We appreciate the reviewer’s positive and constructive feedback. We provide point-by-point responses to each of your comments.
>
>
>
> > **Computational cost and hyperparameters**
>
> Thank you for highlighting this perspective. We have added a detailed comparison of computational costs for our method and all baselines in the appendix. While our approach incurs higher computational cost than methods that do not require neural network training (e.g., Local Ellipsoid, empirical copula, Gaussian copula), we believe the performance gains justify this additional computational cost. Other baselines involving neural networks (e.g., CONTRA, CopulaCPTS), machine learning models (e.g., MultiDimSPCI), or optimal-transport solvers (e.g., OT-CP) exhibit comparable computational cost.
>
> The additional hyperparameters introduced specifically by our method are the covariance scale and the null condition probability. For both hyperparameters, prior works (e.g., about flow matching [1] and classifier-free guidance [2]) provide standard recommended values, and our experiments further confirm that our method performed well using the standard recommended values. Finally, although our experiments focus on datasets with $d_y \leq 8$, we observe that performance remains stable across similar architectural and training configurations, suggesting that our method is robust to these choices.
>
> [1] Flow Matching for Generative Modeling, Lipman et al., ICLR 2023
>
> [2] Classifier-Free Diffusion Guidance, Ho et al., 2021
>
>
>
> > **Potential reason for the performance gain**
>
> Using conditional quantile estimation using machine learning methods such as quantile random forest [1] or Transformer [2] showed successful performance on constructing 1-dimensional prediction interval. However, constructing prediction sets for multi-dimensional outcome requires additional problem to determine geometric form of the sets, which is significantly different from constructing a prediction interval where two points of values are sufficient to construct a prediction interval. Therefore, without significant modification of the model architecture or method, it is difficult to directly apply existing such methods based on machine learning methods for multi-dimensional prediction sets.
>
> Similar to the reviewer’s point, a few existing studies use conditional quantile estimation with pre-defined geometric shape (e.g., ellipsoidal prediction set [3] as we included in baselines). However, our method outperformed the approach. We believe the flexibility of shape without being restricted to a specific geometric shape with good expressivity of neural networks used to model the vector field of guided flow lead to significant gain of our proposed methods compared to such baselines. This can be understood as additional benefit over conditional quantile estimation since we also use encoder that enable to model the conditional quantile. We have briefly added this point of discussion in the method section 3.2.
>
>
> [1] Sequential Predictive Conformal Inference for Time Series, Xu and Xie, ICML 2023
>
> [2] Transformer Conformal Prediction for Time Series, Lee et al., arxiv 2024
>
> [3] Conformal prediction for multi-dimensional time series by ellipsoidal sets, Xu et al., ICML 2024

---

> ### Author Response · Authors · 2025-11-25
> **Rebuttal (2/2)**
>
> > **On the assumptions for theoretical results**
>
> 1. **Assumption about compact domain of features and outcomes.**
>
> The compactness of the feature space and outcome space is not required for our main theoretical guarantees. This assumption is only needed when we explicitly wish to ensure that  $h$ in a compact domain $\mathcal{H}$. Because this may unintentionally give the impression that compactness is a necessary condition for our method, we have moved the assumption to the appendix and clarified that it is only needed when bounding $\mathcal{H}$ is desired. Moreover, assuming compact domains for features and outcomes is reasonable in practice, as real-world data can be bounded or truncated to a compact region without loss of generality.
>
>
> 2. **Assumption about the flow existence.**
>
> This assumption is required to ensure the flow ODE has a unique solution given by the flow, which follows from the Picard–Lindelöf theorem. This assumption is standard in the studies using flow and as discussed in Remark 4.2, they can be readily satisfied in practice by designing the neural network architecture for the vector field to obey the required Lipschitz and smoothness properties.
>
>
>
> 3. **Assumption about non-conformity scores.**
>
> We first would like to emphasize that, with a properly trained model, $e_i$ are i.i.d. because the source distribution is set to an isotropic Gaussian, regardless of $h_i$ ​. Thus, strong mixing or stationarity of $e_i$​ is not a strict requirement for achieving conditional coverage; rather, these conditions capture exceptional cases in which dependence among $e_i$​ may arise indirectly through the temporal dependence structure of $h_i$​.
>
> In general, a time series can exhibit arbitrary dependence or even non-stationarity, while still having strongly mixing (or even i.i.d.) errors [1]. Consequently, assuming $\epsilon$ is stationary and strongly mixing is a mild assumption and commonly made within similar conformal prediction studies on time series [2,3,4,5]. Since $h_i$ is generated from a finite context window using a learned encoder that summarizes past information, it is reasonable to regard the sequence $h_i$ as strongly mixing as well. Under these conditions, the sequence $e_i = \psi(\epsilon_i | h_i)$ inherits strong mixing, and because the marginal distribution of $e_i$​ is the same for all $i$, the sequence is also stationary.
>
> We additionally assume that the distribution function $F_e(u)$ of the non-conformity scores is Lipschitz continuous with constant $L_{T+1} > 0$ and strictly increasing in $u$. Lipschitz continuity ensures that $F_e(u)$ behaves well for theoretical analysis. Strict monotonicity ensures that quantiles of the non-conformity scores are uniquely defined and have positive probability density. Both assumptions are mild and standard in theoretical analyses within the conformal prediction literature for time series [2,3,4,5].
>
>
> [1] Mixing: properties and examples, Doukhan. P., 2012
>
> [2] Sequential Predictive Conformal Inference for Time Series, Xu and Xie, ICML 2023
>
> [3] Conformal prediction interval for dynamic time-series, Xu and Xie, ICML 2021
>
> [4] Conformal Prediction for Time Series with Modern Hopfield Networks, Auer et al., NeurIPS 2023
>
> [5] Kernel-based optimally weighted conformal time-series prediction, Lee et al., ICLR 2025
>
>
>
> 4. **Assumption about the bi-Lipschitz flow**
>
> This assumption is essentially a two-sided extension of Assumption 4.1, ensuring that both the forward and inverse flows are Lipschitz. While Assumption 4.1 guarantees the existence and uniqueness of the flow, they do not, in general, imply that the inverse transformation is Lipschitz. This bi-Lipschitz condition is stronger and less common in the flow literature, but it is introduced specifically to control the inverse mapping of the prediction residuals and to derive finite-sample conditional coverage bounds. To validate the practicality of this assumption, we conducted an ablation study using iResNet in the experiment section  (which enforces bi-Lipschitzness by design) and empirically observed that imposing this constraint did not adversely affect coverage performance.
>
>
>
> 5. **Assumption about the estimation quality of the base predictor.**
>
> Since the conditional coverage bound depends on the estimation quality of the base predictor $\delta_T$, we provide a discussion at the end of the Theory section showing that any estimator with convergence rate $\mathcal{O}(T^{-a})$ for $a > 0$ ensures that the bound converges to $1-\alpha$ for $T \to \infty$. This requirement is mild and can be satisfied by a broad class of estimators commonly used in practice.

---

### Official Review · Reviewer_bXWm · 2025-10-30

**Soundness:** 2
**Presentation:** 2
**Contribution:** 2
**Rating:** 6
**Confidence:** 3

**Summary:**

This paper introduces a method for uncertainty quantification in multidimensional time series forecasting. The approach begins by using any base forecaster to generate point predictions and residuals. It then encodes recent historical data as contextual information to condition a continuous normalizing flow, which maps an isotropic Gaussian source distribution to the situational residual distribution.

**Strengths:**

(1) The paper features a clear structure and rigorous logic, facilitating readers' comprehensive understanding of the methodology.

(2) The paper introduces a multidimensional uncertainty quantification approach centered on conditional continuous flow. By mapping source distributions to residual distributions, it establishes a unified pathway for constructing prediction sets, demonstrating novelty and general applicability.

(3) The paper presents a comprehensive argumentative framework, rigorously defining the assumptions and applicability of marginal coverage and conditional coverage respectively, accompanied by rigorous proofs.

(4)The experimental design encompasses multiple datasets, diverse output dimensions, and various base predictors. Evaluation metrics are clearly defined, and results effectively validate the proposed methodology

**Weaknesses:**

(1) The paper lacks a corresponding overview of methods and a concise algorithmic workflow.

(2) The rationale for selecting the source distribution and scoring metric is not sufficiently discussed. The paper adopts an isotropic Gaussian distribution as its core design without presenting alternative source distributions or concluding on differences in coverage and set size.

(3) The claim of conditional coverage lacks empirical support. While presented as a theoretical contribution, the experiments only report overall coverage and set size without showing coverage performance grouped by scenario or context.

**Questions:**

(1) Is the precise marginal coverage established solely on theoretical grounds? In practice, do flow estimation errors and base predictor errors influence coverage bias? If so, can an upper bound or consistency conclusion be provided?

(2) At a given nominal level, does the constructed forecast set size exhibit near-optimality (minimal volume)? As the sample size approaches infinity, does it converge consistently to the optimal solution? What assumptions or conditions are required?

(3) Are the assumptions for achieving conditional coverage, such as the strong mixing condition, empirically supported in real data? If not satisfied, can a robust adaptation scheme be proposed that accommodates both the data and the proposed method?

(4) If a distribution shift occurs during the testing phase, will the context encoder and conditional stream cause coverage degradation? Does the method adaptively handle distribution shifts while still meeting the specified coverage rate?

---

> ### Author Response · Authors · 2025-11-25
> **Rebuttal (1/2)**
>
> We sincerely appreciate the reviewer’s valuable comments. Below, we provide point-by-point responses to your comments.
>
>
> > **Need for the method overview and algorithm**
>
> We appreciate the reviewer’s thoughtful suggestion. We have added a graphical overview of our proposed method (Figure 1) and algorithm for the training of our method (Algorithm 1) to enhance the readability of the paper.
>
>
> > **Choice of the source distribution**
>
> Thank you for the reviewer’s suggestion. We adopt a Gaussian source distribution because it offers several advantages including (1) it yields analytically simple expressions for the conditional vector field and probability path, and (2) it enables an exact and easily computable target vector field, which substantially simplifies training under flow matching. For these reasons, the Gaussian source distribution has become a standard choice in both flow-based and diffusion models. The Gaussian source also has the benefit in our method that it is straightforward to define and manipulate a set containing the probability mass equal to the target coverage through a ball. While other source distributions can be used in principle, we believe the potential benefits are limited, as the source distribution itself does not play a direct theoretical role in our coverage guarantees.
>
>
> > **Empirical support for conditional coverage**
>
> We agree with the reviewer that average coverage alone does not show empirical evidence of conditional coverage. However, evaluating conditional coverage directly is challenging, as it requires multiple samples sharing the same feature representation—that typically demands carefully curated synthetic data or ad-hoc data engineering strategies when working with real-world datasets.
>
> Given these challenges, we approximate conditional coverage using rolling coverage computed at each time index. Rolling coverage at time index $i$ is defined as the average coverage over a predefined rolling window of $m$ test samples. While rolling coverage does not provide an exact estimate of conditional coverage, its pattern allows us to diagnose whether a method systematically violates conditional coverage.
>
> We have added rolling coverage results on the wind dataset for our proposed method, as well as for the strong baseline methods that exhibited comparable coverage to FCP, in the appendix. These results show that our method consistently achieves conditional coverage close to the target coverage, with variances that are comparable to or smaller than those of the baseline methods.
>
>
> > **Theoretical guarantee justification in marginal coverage**
>
> Thank you for the insightful comment. Lemma 4.3 states that a flow transformation preserves the probability mass of any measurable set. The exact marginal coverage established in Proposition 4.4 can be obtained by applying this property to the ball defined under the source distribution, rather than by transforming residuals back to $x_0$.
>
> While Lemma 4.3 guarantees marginal coverage, we agree with the reviewer that approximation error may exist as the inverse transformation is far away from the source distribution we set, and we acknowledge that theoretical analysis of conditional coverage relies on the assumption of ‘the guided flow provides a sufficiently accurate approximation of the target distribution from the source distribution’. Therefore, we have added this statement in the Proposition 4.4, Theorem 4.13, and Corollary 4.18. Although this assumption is often implicitly made in generative modeling with flows and empirically shown that the assumption holds well with stable target coverages in our experiments, we agree that the assumption warrants further discussion, particularly in light of ongoing research that aims to improve the accuracy of both the terminal and initial conditions of the flow mapping through dynamic optimal transport [1,2]. We have therefore added a discussion of this point in the conclusion.
>
> [1] Improving and generalizing flow-based generative models with minibatch optimal transport, Tong et al., TMLR 2024
>
> [2] Computing high-dimensional optimal transport by flow neural networks, Xu et al., AISTATS 2025

---

> ### Author Response · Authors · 2025-11-25
> **Rebuttal (2/2)**
>
> > **Optimal set size**
>
> To the best of my knowledge, although conformal prediction provides marginal or conditional coverage guarantees, it is not feasible to determine whether the resulting prediction set has minimal volume. This is because the set is constructed by using non-conformity scores without making any assumptions about their underlying distribution.
>
>
> > **Regarding assumptions for conditional coverage bound**
>
>
>
> 1. **Assumption about non-conformity scores.**
>
> We first would like to emphasize that, with a properly trained model, $e_i$ are i.i.d. because the source distribution is set to an isotropic Gaussian, regardless of $h_i$ ​. Thus, strong mixing or stationarity of $e_i$​ is not a strict requirement for achieving conditional coverage; rather, these conditions capture exceptional cases in which dependence among $e_i$​ may arise indirectly through the temporal dependence structure of $h_i$​.
>
> In general, a time series can exhibit arbitrary dependence or even non-stationarity, while still having strongly mixing (or even i.i.d.) errors [1]. Consequently, assuming $\epsilon$ is stationary and strongly mixing is a mild assumption and commonly made within similar conformal prediction studies on time series [2,3,4,5]. Since $h_i$ is generated from a finite context window using a learned encoder that summarizes past information, it is reasonable to regard the sequence $h_i$ as strongly mixing as well. Under these conditions, the sequence $e_i = \psi(\epsilon_i | h_i)$ inherits strong mixing, and because the marginal distribution of $e_i$​ is the same for all $i$, the sequence is also stationary.
>
> We additionally assume that the distribution function $F_e(u)$ of the non-conformity scores is Lipschitz continuous with constant $L_{T+1} > 0$ and strictly increasing in $u$. Lipschitz continuity ensures that $F_e(u)$ behaves well for theoretical analysis. Strict monotonicity ensures that quantiles of the non-conformity scores are uniquely defined and have positive probability density. Both assumptions are mild and standard in theoretical analyses within the conformal prediction literature for time series [2,3,4,5].
>
>
> [1] Mixing: properties and examples, Doukhan. P., 2012
>
> [2] Sequential Predictive Conformal Inference for Time Series, Xu and Xie, ICML 2023
>
> [3] Conformal prediction interval for dynamic time-series, Xu and Xie, ICML 2021
>
> [4] Conformal Prediction for Time Series with Modern Hopfield Networks, Auer et al., NeurIPS 2023
>
> [5] Kernel-based optimally weighted conformal time-series prediction, Lee et al., ICLR 2025
>
>
>
> **2. Assumption about the bi-Lipschitz flow**
>
> This assumption is essentially a two-sided extension of Assumption 4.1, ensuring that both the forward and inverse flows are Lipschitz. While Assumption 4.1 guarantees the existence and uniqueness of the flow, they do not, in general, imply that the inverse transformation is Lipschitz. This bi-Lipschitz condition is stronger and less common in the flow literature, but it is introduced specifically to control the inverse mapping of the prediction residuals and to derive finite-sample conditional coverage bounds. To validate the practicality of this assumption, we conducted an ablation study using iResNet in the experiment section  (which enforces bi-Lipschitzness by design) and empirically observed that imposing this constraint did not adversely affect coverage performance.
>
>
> > **Distributional shift**
>
> Although we did not explicitly include datasets with controlled distributional shifts, we believe the real-world time series used in our experiments naturally exhibit such shifts. This setting provides a reasonable approximation of practical distributional shifts, and our empirical results suggest that the proposed method is at least partially robust to the distributional shift than baselines. In particular, we believe the encoder has enough capacity to learn guidance that adapts distributional shift. Across multiple real-world datasets, our method consistently demonstrates stable coverages, indicating its ability to better handle potential distributional shifts in practice.

---

### Official Review · Reviewer_PND7 · 2025-10-31

**Soundness:** 2
**Presentation:** 2
**Contribution:** 2
**Rating:** 4
**Confidence:** 3

**Summary:**

This paper studies the construction of calibrated prediction intervals for multivariate time-series
data.  Building on Guided Flow, the proposed method transforms residuals
into a Gaussian variable and then derives a confidence region by inverting the
Gaussian distribution. Some theoretical guarantees for the method are provided in the paper,
and the proposed method is evaluated on numerical experiments.

**Strengths:**

1. The paper considers an important yet challenging problem: constructing
calibrated prediction intervals for multivariate time-series data.
2. The proposed method has provided some fresh perspectives on the problem.

**Weaknesses:**

1. Although the goal is to construct prediction intervals, the proposed method does not seem to
be a "conformal prediction" method in the usual sense: it does not leverage any (approximate) exchangeability
of the conformity scores to determine the prediction region.
Instead, it seems closer to a nonparametric method that directly models the residual distribution and then constructs prediction sets.

2. The theoretical guarantees are unclear. For marginal coverage, Proposition 4.6 states that the prediction set produced by the algorithm satisfies coverage guarantee "if the ball $B_{\alpha}$ defining the prediction set in equation (5) has probability mass $1-\alpha$". This is a strong "if", since $B_\alpha$ is derived under Gaussian distribution, while $\hat e(y_i)$ need not be exactly or even close to Gaussian in the presence of approximation error or model misspecification. As written, it is unclear what unconditional coverage guarantee the method provides and under what assumptions (or calibration procedures) the condition is satisfied.

3. The theoretical results rely on numerous assumptions that are not fully justified. A more thorough discussion—motivating each assumption, assessing plausibility in typical applications, and exploring sensitivity to violations—would greatly strengthen the work.

**Questions:**

Please refer to the "Weaknesses" section.

---

> ### Author Response · Authors · 2025-11-25
> **Rebuttal (1/2)**
>
> We are thankful for the reviewer’s valuable comments. Below, we provide point-by-point responses to your comments.
>
>
> > **Concern about the method in the context of conformal prediction**
>
> Thank you for this thoughtful comment. While it is true that classical conformal prediction relies on the exchangeability of non-conformity scores, this assumption is well-known to be violated in time-series settings. As discussed in the introduction, both prior work and our own baselines empirically demonstrate that conformal prediction methods that are based on the exchangeability perform poorly on time series data. This challenge has motivated a growing body of research that replaces exchangeability with direct modeling of the conditional distribution of non-conformity scores, typically through conditional quantile estimation [1–5]. Our method follows this established line of work.
>
> Although these approaches differ from classical CP that are based on the exchangeability, they preserve the core of conformal prediction—using non-conformity scores to define prediction regions—while replacing the exchangeability assumption with a conditional modeling assumption that is more appropriate for the data that may violate the exchangeability. In this sense, the method remains a valid conformal prediction, but adapted to scenarios where exchangeability is violated.
> We also note that similar approaches exist in the time-series forecasting literature that directly model outcomes or impose specific distributional forms [6,7]. However, these methods do not construct prediction sets through non-conformity scores and therefore cannot be considered as a valid conformal prediction.
>
> [1] Sequential Predictive Conformal Inference for Time Series, Xu and Xie, ICML 2023
>
> [2] Conformal prediction for time series with modern hopfield networks, Auer et al., NeurIPS 2023
>
> [3] Conformal prediction for multi-dimensional time series by ellipsoidal sets, Xu et al., ICML 2024
>
> [4] Relational Conformal Prediction for Correlated Time Series, Cini et al., ICML 2025
>
> [5] Transformer Conformal Prediction for Time Series, Lee et al., arxiv 2024
>
> [6] Flow Matching with Gaussian Process Priors for Probabilistic Time Series Forecasting, Kollovieh et al., ICLR 2025
>
> [7] Multivariate Probabilistic Time Series Forecasting via Conditioned Normalizing Flows, Rasul et al., ICLR 2021
>
>
> > **Theoretical guarantee justification in marginal coverage**
>
> Thank you for the insightful comment. Lemma 4.3 states that a flow transformation preserves the probability mass of any measurable set. The exact marginal coverage established in Proposition 4.4 can be obtained by applying this property to the ball defined under the source distribution, rather than by transforming residuals back to $x_0$.
>
> While Lemma 4.3 guarantees marginal coverage, we agree with the reviewer that approximation error may exist as the inverse transformation is far away from the source distribution we set, and we acknowledge that theoretical analysis of conditional coverage relies on the assumption of ‘the guided flow provides a sufficiently accurate approximation of the target distribution from the source distribution’. Therefore, we have added this statement in the Proposition 4.4, Theorem 4.13, and Corollary 4.18. Although this assumption is often implicitly made in generative modeling with flows and empirically shown that the assumption holds well with stable target coverages in our experiments, we agree that the assumption warrants further discussion, particularly in light of ongoing research that aims to improve the accuracy of both the terminal and initial conditions of the flow mapping through dynamic optimal transport [1,2]. We have therefore added a discussion of this point in the conclusion.
>
>
> [1] Improving and generalizing flow-based generative models with minibatch optimal transport, Tong et al., TMLR 2024
>
> [2] Computing high-dimensional optimal transport by flow neural networks, Xu et al., AISTATS 2025

---

> ### Author Response · Authors · 2025-11-25
> **Rebuttal (2/2)**
>
> > **On the assumptions for theoretical results**
>
> 1. **Assumption about compact domain of features and outcomes.**
>
> The compactness of the feature space and outcome space is not required for our main theoretical guarantees. This assumption is only needed when we explicitly wish to ensure that  $h$ in a compact domain $\mathcal{H}$. Because this may unintentionally give the impression that compactness is a necessary condition for our method, we have moved the assumption to the appendix and clarified that it is only needed when bounding $\mathcal{H}$ is desired. Moreover, assuming compact domains for features and outcomes is reasonable in practice, as real-world data can be bounded or truncated to a compact region without loss of generality.
>
>
> 2. **Assumption about the flow existence.**
>
> This assumption is required to ensure the flow ODE has a unique solution given by the flow, which follows from the Picard–Lindelöf theorem. This assumption is standard in the studies using flow and as discussed in Remark 4.2, they can be readily satisfied in practice by designing the neural network architecture for the vector field to obey the required Lipschitz and smoothness properties.
>
>
>
> 3. **Assumption about non-conformity scores.**
>
> We first would like to emphasize that, with a properly trained model, $e_i$ are i.i.d. because the source distribution is set to an isotropic Gaussian, regardless of $h_i$ ​. Thus, strong mixing or stationarity of $e_i$​ is not a strict requirement for achieving conditional coverage; rather, these conditions capture exceptional cases in which dependence among $e_i$​ may arise indirectly through the temporal dependence structure of $h_i$​.
>
> In general, a time series can exhibit arbitrary dependence or even non-stationarity, while still having strongly mixing (or even i.i.d.) errors [1]. Consequently, assuming $\epsilon$ is stationary and strongly mixing is a mild assumption and commonly made within similar conformal prediction studies on time series [2,3,4,5]. Since $h_i$ is generated from a finite context window using a learned encoder that summarizes past information, it is reasonable to regard the sequence $h_i$ as strongly mixing as well. Under these conditions, the sequence $e_i = \psi(\epsilon_i | h_i)$ inherits strong mixing, and because the marginal distribution of $e_i$​ is the same for all $i$, the sequence is also stationary.
>
> We additionally assume that the distribution function $F_e(u)$ of the non-conformity scores is Lipschitz continuous with constant $L_{T+1} > 0$ and strictly increasing in $u$. Lipschitz continuity ensures that $F_e(u)$ behaves well for theoretical analysis. Strict monotonicity ensures that quantiles of the non-conformity scores are uniquely defined and have positive probability density. Both assumptions are mild and standard in theoretical analyses within the conformal prediction literature for time series [2,3,4,5].
>
>
> [1] Mixing: properties and examples, Doukhan. P., 2012
>
> [2] Sequential Predictive Conformal Inference for Time Series, Xu and Xie, ICML 2023
>
> [3] Conformal prediction interval for dynamic time-series, Xu and Xie, ICML 2021
>
> [4] Conformal Prediction for Time Series with Modern Hopfield Networks, Auer et al., NeurIPS 2023
>
> [5] Kernel-based optimally weighted conformal time-series prediction, Lee et al., ICLR 2025
>
>
>
> 4. **Assumption about the bi-Lipschitz flow**
>
> This assumption is essentially a two-sided extension of Assumption 4.1, ensuring that both the forward and inverse flows are Lipschitz. While Assumption 4.1 guarantees the existence and uniqueness of the flow, they do not, in general, imply that the inverse transformation is Lipschitz. This bi-Lipschitz condition is stronger and less common in the flow literature, but it is introduced specifically to control the inverse mapping of the prediction residuals and to derive finite-sample conditional coverage bounds. To validate the practicality of this assumption, we conducted an ablation study using iResNet in the experiment section  (which enforces bi-Lipschitzness by design) and empirically observed that imposing this constraint did not adversely affect coverage performance.
>
>
>
> 5. **Assumption about the estimation quality of the base predictor.**
>
> Since the conditional coverage bound depends on the estimation quality of the base predictor $\delta_T$, we provide a discussion at the end of the Theory section showing that any estimator with convergence rate $\mathcal{O}(T^{-a})$ for $a > 0$ ensures that the bound converges to $1-\alpha$ for $T \to \infty$. This requirement is mild and can be satisfied by a broad class of estimators commonly used in practice.

---

### Official Review · Reviewer_s5Xm · 2025-11-01

**Soundness:** 2
**Presentation:** 3
**Contribution:** 1
**Rating:** 0
**Confidence:** 4

**Summary:**

The proposed work presents a novel conformal production method for multidimensional time series by combining a transformer and a flow model. The obtained prediction region incorporates the historical context of the time series and is based on the transformation of residuals, creating an Euclidean ball in the latent space of flow residuals.  The efficacy of the proposed method is evaluated across several settings and various datasets.

**Strengths:**

The experiments are conducted with multiple base predictors, and various baselines have been compared against, with FCP coming out as the best in the experiments.

The experiments are repeated across several runs, providing uncertainty statements that give a better picture of the method's efficacy.

**Weaknesses:**

To best judge a conformal method, it is imperative to compare its performance across different significance levels, and plotting a calibration curve is particularly helpful.

Most of the theoretical results are well-established or fundamental.

See the "Questions" below for more.

**Questions:**

1. Line 016, "leveraging correlations in features and non-conformity scores" Did the authors want to allude to the exchangeability assumption in the traditional CP setup?

2. The proposal is to use a flow with Classifier-free guidance; however, the idea only requires a kind of flow-based model that can transform a given distribution to another and is bijective. This makes the idea somewhat restrictive, given that a practitioner might want to use the Diffusion Model, classifier-based guidance, or a discrete flow.

3. The idea to model the residual distribution is already present in the literature. For example, Res-CONTRA [1] transforms the residuals as a means of learning the calibration, while also utilising an Euclidean ball to define conformity scores.

4. Line 045: If I am not wrong, the cited Barber et al 2023 does not require the exchangeability assumption; I believe this is a mis-citation.

5. Line 077: "Despite these efforts, existing methods remain limited to univariate outcomes or assume access to multiple i.i.d. time series." This statement seems misleading, given that many methods nowadays discuss multivariate outcomes or single time series setups.

6. Line 095: There are more works apart from Xu et. al 2024 that also discuss single time series settings with multiple step-ahead forecasting. CAFHT, JANET [2, 3] is one such example. Note that their work is also applicable to multiple trajectories. More importantly, any idea, such as ACI, can incorporate multivariate responses by simply using a non-conformity score that caters to multivariate responses.

7. Line 113: with y_i = f(x_i), it looks like there is no dependence on history y_{i-1}. Is that intentional?

8. Line 119: Similarly,  z_i seems only to include x. I mean to say there is a lack of clarity around the notations.

9. The idea of using a ball is already present in CONTRA, such as in Eq. 7. Furthermore, they have a computationally efficient version, as one only needs to work around the boundary of the ball.

10. One issue with using Flow-matching is solving ODE or computing the divergence for the area computation. From this perspective, it seems more natural to use discrete flows.

11. Theorem B.4, Lemma 4.5, and Proposition 4.6 present a similar result to one in CONTRA.

12. Conditional coverage results, while important, depend on strong assumptions such as strongly mixing.

13. Line 373-377: I am confused about the data splits here. Why is it that there is no calibration set needed for FCP? Furthermore, using the training and the validation sets for calibration skews the coverage guarantees, as far as I can understand.

14. Line 385: The target confidence is set as high as 0.95, which is okay. But it is necessary to show the performance of the proposed method with different significance levels. A calibration curve might be helpful to see if the proposed method works well in all cases.

15. The theoretical statements are there for conditional coverage, but there is no empirical evidence for the same.


[1] CONTRA: Conformal Prediction Region via Normalizing Flow Transformation Zhenhan FANG · Aixin Tan · Jian Huang
[2] Conformalized Adaptive Forecasting of Heterogeneous Trajectories Yanfei Zhou, Lars Lindemann, Matteo Sesia
[3] JANET: Joint Adaptive predictioN-region Estimation for Time-series Eshant English, Eliot Wong-Toi, Matteo Fontana, Stephan Mandt, Padhraic Smyth, Christoph Lippert

---

> ### Author Response · Authors · 2025-11-25
> **Rebuttal (1/4)**
>
> We appreciate the reviewer’s careful and constructive feedback. We provide point-by-point responses to each of your comments.
>
>
> > **Evaluation across different significance levels**
>
> We agree with the reviewer that evaluating the proposed method and baselines across different significance levels is important. Accordingly, we have added results at the 0.9 confidence level. While a full calibration curve over multiple significance levels would provide a more comprehensive assessment, we updated the results at 0.9 confidence level the appendix, given the limited time available during the rebuttal period.
>
>
>
> > **Novelty of our method and theoretical results compared to existing work, CONTRA**
>
> We acknowledge the reviewer’s concern that some of our theoretical results may appear similar to those in CONTRA [1]. While both methods use normalizing flows as a core component, our approach is fundamentally different from CONTRA in several key aspects:
>
> 1. **Continuous-time vs. discrete-time flows**
>  CONTRA builds on a discrete-time normalizing flow, whereas our method is based on a continuous-time flow. Although the two share some theoretical background, the continuous-time formulation leads to a substantially different framework, particularly in establishing coverage guarantees.
>
>
> 2. **On the exchangeability assumption**
>  CONTRA relies on the exchangeability, an assumption incompatible with time series data. In contrast, our method explicitly targets time series setting, by overcoming the exchangeability assumption. As a result, the construction of prediction sets differs significantly: CONTRA uses empirical quantiles of transformed residuals, while our method constructs prediction sets at each time step by transforming a probability-mass–containing ball through a guided flow conditioned on contextual information.
>
>
> 3. **Conditional coverage consideration**
>  While we provided conditional coverage bound, CONTRA does not provide any conditional coverage guarantees or bounds.
>
>
> **We believe that the conceptual similarity between our method and CONTRA [1] may have contributed to the negative impression of our work. While both methods share a high-level idea, CONTRA and our method differ fundamentally in several aspects, including those mentioned above. We provide detailed answers to your specific questions below and we are also happy to address any additional questions and clarify these distinctions further.**
>
>
> [1] CONTRA: Conformal Prediction Region via Normalizing Flow Transformation, Fang et al., ICLR 2025
>
>
> > **The proposal is to use a flow with Classifier-free guidance; however, the idea only requires a kind of flow-based model that can transform a given distribution to another and is bijective. This makes the idea somewhat restrictive ..**
>
>
> We appreciate this suggestion and partially agree with the concern. However, we would like to clarify several important points:
>
> 1. **Classifier-free guidance is not a limitation in our setting.**
> We use CFG because classifier-based guidance is not feasible for time-series prediction. Classifier-based guidance requires learning a time-dependent classifier, but in time-series prediction, the contextual variable $h$ is not explicitly available in the data. Therefore, there is no supervised signal available to train such a classifier. CFG thus serves as a practical and principled mechanism to condition the flow on past information without requiring an auxiliary classifier.
> Although our task differs from image generation, prior work [1,2] has shown that CFG often outperforms classifier-based guidance while also being more stable and easier to train. For these reasons, CFG is generally favored in practice rather than being considered a restrictive choice.
>
>
> 2. **On the diffusion models and discrete flows.**
> As the reviewer correctly pointed out, we relies on the bijective property of flow for coverage guarantee, so it cannot be straightforward to use diffusion models. However, this is because of the fundamental property of flow and diffusion model, not because of CFG or our proposed method’s restricted capacity. While we think applying diffusion to conformal prediction is interesting future work, we believe considering diffusion requires redesgin fundamentally therefore out of scope of this work. Importantly, neither CFG nor our method precludes the use of discrete flows. In fact, discrete-flow architectures are fully compatible with our framework (such as [3] has already used CFG within discrete flow models).
>
>
> [1] Classifier-Free Diffusion Guidance, Ho and Salimans, 2022
>
> [2] High-Resolution Image Synthesis with Latent Diffusion Models, Rombach, CVPR 2022
>
> [3] Normalizing Flows are Capable Generative Models, Zhai et al., ICML 2025

---

> ### Author Response · Authors · 2025-11-25
> **Rebuttal (2/4)**
>
> > **Line 016, Did the authors want to allude to the exchangeability assumption in the traditional CP setup?**
>
> Yes, the reviewer’s understanding is correct. We aimed to overcome the exchangeability assumption by ‘leveraging correlations in features and non-conformity scores’ to estimate non-conformity scores conditionally. I think it will be more clear to mention ‘to overcome the exchangeability assumption’ therefore we have added it.
>
>
> > **Line 045: mis-citation about Barber et al 2023 et al.**
>
>
> Thank you for pointing this out. We know that (Barber et al. 2023) did not assume the exchangeability in their method as we cited in line 53. We wanted to support the claim ‘most existing CP methods rely on the assumption of data exchangeability’ by referencing (Barber et al. 2023). While its not a wrong citation, we admit that it could be confusing, so we removed (Barber et al. 2023) citation from line 45. We believe the claim is well-supported without the citation.
>
>
> > **Line 077: "Despite these efforts, existing methods remain limited to univariate outcomes or assume access to multiple i.i.d. time series." This statement seems misleading, given that many methods nowadays discuss multivariate outcomes or single time series setups.**
>
>
> We acknowledge the reviewer’s point. Our original intention was to emphasize that comparatively fewer works address conformal prediction for time series with multi-dimensional outcomes, as opposed to (1) univariate time-series settings or (2) settings that assume access to multiple i.i.d. time series. However, we agree that the way this was phrased could be interpreted as misleading and redundant. Therefore, we have removed this statement from the paragraph.
>
>
>
> > **Line 095: There are more works apart from Xu et. al 2024 that also discuss single time series settings with multiple step-ahead forecasting. CAFHT, JANET [1, 2] is one such example ...**
>
> Thank you for the opportunity to clarify this point. We believe there may be a misunderstanding regarding the problem settings considered in the cited works.
>
> 1. Xu et al. (2024) [3] did not study the setting of a single time series with multi-step forecasting. Instead, they focused on single-step forecasting of multivariate outcomes under a single time series, which is exactly the same setting as ours. Therefore, their work is directly comparable and aligned with our problem formulation.
> Regarding the reviewer’s comment that “any idea that can incorporate multivariate responses by simply using a multivariate non-conformity score” may apply broadly: while this is true in principle, the specific studies mentioned by the reviewer address a different problem from ours.
>
> 2. CAFHT [1] does not consider a single-time-series setting. Their method explicitly assumes multiple i.i.d. time series, which is stated in their introduction, problem formulation, and experimental setup. Their experiments also use datasets with many independent trajectories (e.g., 500 synthetic trajectories and 291 pedestrian trajectories).
>
>
> 3. JANET [2] similarly assumes the availability of multiple time series where each of time series is univariate, as stated in their method section: “we assume that the time series are univariate… in the case of a single time series, we treat the permutations of the single series as distinct, exchangeable series.”. Moreover, all their single time series experiments are restricted to univariate outcomes, consistent with their stated assumptions.
>
>
> In contrast, our study directly addresses multivariate outcome under a single observed time series without assuming multiple i.i.d. trajectories or permutation-based augmentation.
>
> [1] Conformalized Adaptive Forecasting of Heterogeneous Trajectories, Zhou et al., ICML 2024
>
> [2] JANET: Joint Adaptive predictioN-region Estimation for Time-series, English et al., Machine Learning 2024
>
> [3] Conformal prediction for multi-dimensional time series by ellipsoidal sets, Xu et al., ICML 2024
>
>
> > **Clarification on $y_i = f(x_i)$ and $z_i$**
>
> We appreciate for the reviewer that pointing this out. It is not intentional and the input for the $\hat{f}$ should be the context window of features depending on the point predictor. We corrected the notation to $\hat{y_i}$ = $\hat{f}(x_{(i-k):i})$. We corrected $z_i$ as ‘we construct $z_i$ by concatenating the past $w$ features and prediction residuals’ throughout the manuscript for clear clarification.

---

> ### Author Response · Authors · 2025-11-25
> **Rebuttal (3/4)**
>
> > **The idea of using a ball is already present in CONTRA, such as in Eq. 7. Furthermore, they have a computationally efficient version, as one only needs to work around the boundary of the ball.**
>
>
> Thank you for pointing this out. We agree that the idea of using a ball is also used in CONTRA [1]. However, there are fundamental differences in constructing prediction sets by using the ball. In CONTRA, the ball is defined using calibration sets by empirical quantile of the transformed samples (i.e., outcomes in CONTRA and prediction residuals in ResCONTRA). Their usage of defining a ball is to obtain the radius of the ball corresponding to the empirical quantile of the transformed samples. However, our method uses a ball in different way, as a set that contains the probability mass that is equal amount of the target coverage, so that the transformed set for prediction residuals conditioned on the guidance also contain the same probability mass. While the usage of a ball may look the two methods similar, the two methods are fundamentally different. This is clear by looking at CONTRA’s marginal coverage guarantee, where they did not use the property of flow.
>
>
> As the reviewer noted, CONTRA uses only the boundary of the prediction set in displaying the prediction set. However, this approach is not specific to CONTRA—it can also be applied to our method, as well as other transformation-based methods such as OT-CP [2]. In fact, we used the same boundary-only visualization approach for our 2-dimensional examples shown in Figure 2. Importantly, this should not be interpreted as a computationally efficient version of the method itself. Transforming only the boundary of a prediction set does not provide information about either the size or coverage of the set; it is simply a convenient and efficient visualization technique when working in two dimensions.
>
> [1] CONTRA: Conformal Prediction Region via Normalizing Flow Transformation, Fang et al., ICLR 2025
>
> [2] Optimal Transport-based Conformal Prediction, Thurin et al., ICML 2025
>
>
>
> > **One issue with using Flow-matching is solving ODE or computing the divergence for the area computation. From this perspective, it seems more natural to use discrete flows.**
>
> We agree that, specifically for computing areas, evaluating the divergence can be more computationally expensive than computing the Jacobian determinant in a discrete-time flow. However, area computation is only one aspect of conformal prediction, and thus it is important to view this through the lens of a broader trade-off.
>
> Continuous-time flows offer advantages as well—for example, they are generally easier to train using flow matching compared to discrete flows. In practice, both modeling approaches involve benefits and drawbacks across dimensions such as training stability, computational efficiency, flexibility, and empirical performance. A comprehensive comparison that accounts for these factors would be needed to determine which approach is more “natural” for conformal prediction, and we consider such an analysis to be valuable future work.
>
>
> > **Theorem B.4, Lemma 4.5, and Proposition 4.6 present a similar result to one in CONTRA.**
>
> Lemma 4.5 and Proposition 4.6 are fundamentally different from the marginal coverage guarantee in CONTRA [1], both in their assumptions and in their theoretical underpinnings. In CONTRA, marginal coverage is established via a quantile-inflation lemma that relies on i.i.d. data, where the normalizing flow serves as a bijective mapping between the outcome (i.e., transformed residuals in ResCONTRA) and samples from a source distribution.
>
> In contrast, our marginal coverage guarantee (Proposition 4.6) is derived from a completely different principle: the measure-preserving property of diffeomorphisms. Lemma 4.5 states that a flow transformation preserves the probability mass of any measurable set, and this property is central to our proof. This type of argument does not hold in CONTRA’s setting and is not discussed in their analysis.
>
> Therefore, although both works aim to establish marginal coverage, the theoretical mechanisms, assumptions, and proof techniques used in Proposition 4.6 (via Lemma 4.5) are fundamentally distinct from those employed in CONTRA.
> Theorem B.4 is a fundamental topological theorem stating that a closed and connected set remains closed and connected under a continuous mapping. This property is general and can be applied broadly.
>
> [1] CONTRA: Conformal Prediction Region via Normalizing Flow Transformation, Fang et al., ICLR 2025

---

> ### Author Response · Authors · 2025-11-25
> **Rebuttal (4/4)**
>
> > **Conditional coverage results, while important, depend on strong assumptions such as strongly mixing.**
>
> We first would like to emphasize that, with a properly trained model, $e_i$ are i.i.d. because the source distribution is set to an isotropic Gaussian, regardless of $h_i$ ​. Thus, strong mixing or stationarity of $e_i$​ is not a strict requirement for achieving conditional coverage; rather, these conditions capture exceptional cases in which dependence among $e_i$​ may arise indirectly through the temporal dependence structure of $h_i$​.
>
> In general, a time series can exhibit arbitrary dependence or even non-stationarity, while still having strongly mixing (or even i.i.d.) errors [1]. Consequently, assuming $\epsilon$ is stationary and strongly mixing is a mild assumption and commonly made within similar conformal prediction studies on time series [2,3,4,5]. Since $h_i$ is generated from a finite context window using a learned encoder that summarizes past information, it is reasonable to regard the sequence $h_i$ as strongly mixing as well. Under these conditions, the sequence $e_i = \psi(\epsilon_i | h_i)$ inherits strong mixing, and because the marginal distribution of $e_i$​ is the same for all $i$, the sequence is also stationary.
>
>
> [1] Mixing: properties and examples, Doukhan. P., 2012
>
> [2] Sequential Predictive Conformal Inference for Time Series, Xu and Xie, ICML 2023
>
> [3] Conformal prediction interval for dynamic time-series, Xu and Xie, ICML 2021
>
> [4] Conformal Prediction for Time Series with Modern Hopfield Networks, Auer et al., NeurIPS 2023
>
> [5] Kernel-based optimally weighted conformal time-series prediction, Lee et al., ICLR 2025
>
>
> > **Line 373-377: I am confused about the data splits here. Why is it that there is no calibration set needed for FCP? Furthermore, using the training and the validation sets for calibration skews the coverage guarantees, as far as I can understand.**
>
> Since our method directly models the conditional distribution of the prediction residual at each time step, it does not require a calibration step. As a result, we use the training set solely for model fitting and a validation set for early stopping. We note that this setup is consistent with a growing body of work overcoming the exchangeability in time series by modeling the conditional distribution of non-conformity scores [1–5].
>
> [1] Sequential Predictive Conformal Inference for Time Series, Xu and Xie, ICML 2023
>
> [2] Conformal prediction for time series with modern hopfield networks, Auer et al., NeurIPS 2023
>
> [3] Conformal prediction for multi-dimensional time series by ellipsoidal sets, Xu et al., ICML 2024
>
> [4] Relational Conformal Prediction for Correlated Time Series, Cini et al., ICML 2025
>
> [5] Transformer Conformal Prediction for Time Series, Lee et al., arxiv 2024
>
>
> > **The theoretical statements are there for conditional coverage, but there is no empirical evidence for the same.**
>
> We agree with the reviewer that average coverage alone does not show empirical evidence of conditional coverage. However, evaluating conditional coverage directly is challenging, as it requires multiple samples sharing the same feature representation—that typically demands carefully curated synthetic data or ad-hoc data engineering strategies when working with real-world datasets.
>
> Given these challenges, we approximate conditional coverage using rolling coverage computed at each time index. Rolling coverage at time index $i$ is defined as the average coverage over a predefined rolling window of $m$ test samples. While rolling coverage does not provide an exact estimate of conditional coverage, its pattern allows us to diagnose whether a method systematically violates conditional coverage.
>
> We have added rolling coverage results on the wind dataset for our proposed method, as well as for the strong baseline methods that exhibited comparable coverage to FCP, in the appendix. These results show that our method consistently achieves conditional coverage close to the target coverage, with variances that are comparable to or smaller than those of the baseline methods.

---

> > ### Comment · Reviewer_s5Xm · 2025-11-28
> >
> > Thanks for writing the rebuttal. I have some follow-up questions:
> >
> > 1. "The continuous-time formulation leads to a substantially different framework, particularly in establishing coverage guarantees." - Could you elaborate on how the continuous-time formulation leads to a difference in establishing coverage guarantees?
> >
> > 2. "As the reviewer correctly pointed out, we rely on the bijective property of flow for coverage guarantee, so it cannot be straightforward to use diffusion models." - Would that still be the case if one is using probability flow solutions from the diffusion models that leverage ODE mappings?
> >
> > 3. Related works:
> >    a. Xu et al. (2024) : Indeed, they discussed a single-step setting, but if I understood clearly, the adaptation is straightforward with norm-based conformity scores.
> >    b. CAFHT: You are right about their setting. They blend ideas to learn from a single series and then apply them to different trajectories. However, their approach could be applied seamlessly to a single time series as they are running the ACI [1] algorithm. Also, they mentioned the PID [2] algorithm for time series.
> >   c. JANET: Their single-time series setting seems to be applicable here; they have conformity scores that map to multiple responses.
> >
> > 4. "However, our method uses a ball in different way, as a set that contains the probability mass that is equal amount of the target coverage, so that the transformed set for prediction residuals conditioned on the guidance also contain the same probability mass.": Thanks for clarifying this point. This indeed sets the method apart from CONTRA, which was my main concern. There are still similarities between the two, but they are NOT the same. I will update my score in lieu of this.
> >
> > 5. I see the difference better now. However, the method no longer appears to be a conformal method to me. The idea of calibration was an important way to get conformal guarantees at least in the exchangeable setting, but now I am not sure how your method achieves the marginal coverage. Simply, if the latent distribution is not perfectly modelled to be Gaussian, would the area enclosed by the ball be the same as the method suggests?
> >
> > 6. "Continuous-time flows offer advantages as well—for example, they are generally easier to train using flow matching compared to discrete flows." : I do not agree with this statement. For non-image data, training discrete versions can be equally easy to train, if not easier.
> >
> > 7. Finally, I do not suggest framing the proposed method as a conformal method. It appears very different from what I understand CP methods are.
> >
> >
> >
> > [1] I. Gibbs and E. Cand`es. “Adaptive conformal inference under distribution shift”. In: Adv. Neural Inf. Process. Syst. 34 (2021), pp. 1660–1672.
> > [2] A. N. Angelopoulos, E. J. Cand`es, and R. J. Tibshirani. “Conformal PID control for time series prediction”. In: arXiv preprint arXiv:2307.16895 (2023)

---

> > > ### Author Response · Authors · 2025-12-01
> > > **Follow-up rebuttal (2/3)**
> > >
> > > > *"I see the difference better now. However, the method no longer appears to be a conformal method to me. The idea of calibration was an important way to get conformal guarantees at least in the exchangeable setting…*"
> > >
> > > We would like to clarify that calibration in conformal prediction does not intrinsically require exchangeability. Rather, calibration should be viewed as the general procedure of estimating the desired quantiles of a non-conformity score in order to construct prediction sets. Different conformal prediction methods implement this calibration step differently depending on their assumptions and problem settings. In particular, under time-series or other non-exchangeable settings, training conditional quantile estimators are often used as an alternative calibration step [1-5]. In this sense, our method still performs a calibration step, in a form adapted to overcome the exchangeability assumption.
> > >
> > >
> > > [1] Sequential Predictive Conformal Inference for Time Series, Xu and Xie, ICML 2023
> > >
> > > [2] Conformal prediction for time series with modern hopfield networks, Auer et al., NeurIPS 2023
> > >
> > > [3] Conformal prediction for multi-dimensional time series by ellipsoidal sets, Xu et al., ICML 2024
> > >
> > > [4] Relational Conformal Prediction for Correlated Time Series, Cini et al., ICML 2025
> > >
> > > [5] Transformer Conformal Prediction for Time Series, Lee et al., arxiv 2024
> > >
> > >
> > > > *"but now I am not sure how your method achieves the marginal coverage. Simply, if the latent distribution is not perfectly modelled to be Gaussian, would the area enclosed by the ball be the same as the method suggests?*"
> > >
> > > Thank you for pointing this out. Lemma 4.3 states that a flow transformation preserves the probability mass of any measurable set. The exact marginal coverage established in Proposition 4.4 can be obtained by applying this property to the ball defined under the source distribution, rather than by transforming residuals back to $x_0$, which we did not need to concern about the marginal coverage affected by the approximation error of the source distribution.
> > >
> > > While Lemma 4.3 guarantees marginal coverage, we agree with the reviewer that approximation error may exist as the inverse transformation is far away from the source distribution we set, and we acknowledge that theoretical analysis of conditional coverage relies on the assumption of ‘the guided flow provides a sufficiently accurate approximation of the target distribution from the source distribution’. Although this assumption is often implicitly made in generative modeling with flows and empirically shown that the assumption holds well with stable target coverages in our experiments, we agree that the assumption warrants further discussion, particularly in light of ongoing research that aims to improve the accuracy of both the terminal and initial conditions of the flow mapping through dynamic optimal transport [1,2]. We have therefore added a discussion of this point in the conclusion.
> > >
> > > [1] Improving and generalizing flow-based generative models with minibatch optimal transport, Tong et al., TMLR 2024
> > >
> > > [2] Computing high-dimensional optimal transport by flow neural networks, Xu et al., AISTATS 2025
> > >
> > >
> > > > *""Continuous-time flows offer advantages as well—for example, they are generally easier to train using flow matching compared to discrete flows." : I do not agree with this statement. For non-image data, training discrete versions can be equally easy to train, if not easier.*"
> > >
> > > Thank you for the viewpoint. Our use of “generally easier” was intended in the sense that discrete flows typically rely on likelihood-based training, which can be more challenging to train compared to the flow matching objective. We agree that for non-image datasets, discrete flows can be equally easy to train in practice. However, in high-dimensional settings such as images, discrete flows may require additional care to maintain numerical stability due to the deep composition of transformations. While we do not disagree with the reviewer’s point, we believe it is still reasonable to state that, in general, continuous-time flows trained via flow matching can offer a more straightforward training objective than likelihood-based training of discrete-time flows.

---

> > > ### Author Response · Authors · 2025-12-01
> > > **Follow-up rebuttal (3/3)**
> > >
> > > > *"Finally, I do not suggest framing the proposed method as a conformal method. It appears very different from what I understand CP methods are.*"
> > >
> > > We acknowledge the reviewer’s viewpoint. However, as we discussed in the response above, we believe our method can be classified as a conformal prediction method where the calibration step is conducted by a conditional quantile estimator [1-5]. While our method may appear different due to the use of guided flows to transform the nonconformity scores, this transformation can be understood as a mechanism for more effectively modeling the conditional quantiles of the prediction residuals. Such a transformation is unnecessary for existing conformal methods targeting one-dimensional intervals or those that directly reduce multidimensional outcomes to scalar nonconformity scores in $\mathbb{R}$. Furthermore, as highlighted in our literature review, recent work has explored related ideas—such as mapping nonconformity scores to a reference distribution using optimal transport [6,7]—and has demonstrated strong empirical performance. For these reasons, we believe it is reasonable to consider our method as a valid conformal prediction method.
> > >
> > > [1] Sequential Predictive Conformal Inference for Time Series, Xu and Xie, ICML 2023
> > >
> > > [2] Conformal prediction for time series with modern hopfield networks, Auer et al., NeurIPS 2023
> > >
> > > [3] Conformal prediction for multi-dimensional time series by ellipsoidal sets, Xu et al., ICML 2024
> > >
> > > [4] Relational Conformal Prediction for Correlated Time Series, Cini et al., ICML 2025
> > >
> > > [5] Transformer Conformal Prediction for Time Series, Lee et al., arxiv 2024
> > >
> > > [6] Multivariate conformal prediction using optimal transport. Klein et al., arxiv 2025
> > >
> > > [7] Optimal transport-based conformal prediction, ICML 2025

---

> ### Author Response · Authors · 2025-12-01
> **Follow-up rebuttal (1/3)**
>
> We are glad that our responses were helpful in addressing the reviewer’s concerns. Below we provide point-by-point answers to your follow-up questions:
>
>
> > *"Thanks for clarifying this point. This indeed sets the method apart from CONTRA, which was my main concern. There are still similarities between the two, but they are NOT the same. I will update my score in lieu of this.*"
>
> We are delighted that our response was helpful to address your main concern.
>
>
> > *"Could you elaborate on how the continuous-time formulation leads to a difference in establishing coverage guarantees?*"
>
>
> We would like to emphasize that coverage guarantees can also be established for discrete-time flows, but the formulation details should differ from the continuous-time flow. In the continuous-time setting, one designs a vector field that satisfies Assumption 4.1 for marginal coverage and Assumption 4.6 for conditional coverage, which in turn ensures that the flow ODE satisfies the same assumptions (see Lemma A.7 and Lemma A.8). In contrast, in the discrete-time formulation each transformation $T_1$ must be constructed so that it satisfies these assumptions individually, ensuring that their composition $T_1 \cdot … T_K$ also inherits the required properties.
>
>
> > *"Would that still be the case if one is using probability flow solutions from the diffusion models that leverage ODE mappings?*"
>
> We appreciate the discussion. We also think probability flow can be used in a similar formulation via probability flow ODE [1], which can guarantee the marginal coverage. However, to obtain conditional coverage, we additionally need to control the inverse transformation. In our method, this is achieved by imposing a bi-Lipschitz assumption on the flow, but enforcing an analogous bi-Lipschitz property for probability flow ODEs is not straightforward due to the additional score term. Although this extension is technically challenging, we believe that the application of diffusion with suitable coverage guarantees could yield prediction sets that are comparable to, or even better than, those obtained from flow-based methods. We therefore view this as an interesting direction for future work.
>
> [1] Score-Based Generative Modeling through Stochastic Differential Equations, Song et al., ICLR 2021
>
>
> > *"Related works*"
>
> Xu et al. [1] construct prediction sets using ellipsoids and take the ellipsoidal radius as the nonconformity score, estimating its conditional quantile at each time step using a quantile random forest. Although the specific modeling tools differ, the overall approach is closely aligned with ours: both approaches aim to learn the conditional quantile of prediction residuals and use these quantiles to form prediction sets.
>
>
> We agree that, as the reviewer notes, CAFHT “blends ideas to learn from a single series and then apply them to different trajectories.” However, the prediction bands in CAFHT are constructed from empirical quantiles computed across multiple exchangeable trajectories (see Eqs. 3 and 4 in [2]). When applied to a single time-series setting, this procedure reduces to the original ACI method [3]. PID [4] follows the same principle, simply replacing the ACI interval with the PID interval in Eq. 3 of [2]. For this reason, we emphasize that while ACI itself can be applied in our setting, CAFHT necessarily relies on exchangeability across trajectories and therefore falls outside our problem definition.
>
>
> In [5], it is true that the authors use conformity scores capable of handling multidimensional outcomes (e.g., by taking the maximum across dimensions). However, our point concerns a different aspect: in the single–time-series setting, they treat permutations of the same series as distinct, exchangeable trajectories. This effectively augments a single series into multiple series under an exchangeability assumption. Such an assumption falls outside our problem definition, as we do not treat permutations of a single time series as distinct, exchangeable sequences. Therefore, the applicability of their approach relies on an assumption that is incompatible with our setting.
>
> [1] Conformal prediction for multi-dimensional time series by ellipsoidal sets, Xu et al., ICML 2024
>
> [2] Conformalized Adaptive Forecasting of Heterogeneous Trajectories, Zhou et al., ICML 2024
>
> [3] Adaptive conformal inference under distribution shift,  Gibbs and Candes, NeurIPS 2021
>
> [4] Conformal PID control for time series prediction, Angelopoulos et al., NeurIPS 2023
>
> [5] JANET: Joint Adaptive predictioN-region Estimation for Time-series, English et al., Machine Learning 2024

---

### Author Response · Authors · 2025-11-27
**Rebuttal Summary**

We sincerely thank all reviewers for the valuable feedback. We addressed all questions and concerns point-by-point in the following responses. We have also made several changes and additions to the paper, which are summarized below:


**Evaluation at 0.9 confidence level**

It is important to evaluate the proposed method and baselines on different significance levels to compare the performances and its practically useful for audiences. Therefore, we added evaluation on 0.9 confidence level in the appendix. The overall results remain consistent with those at the 0.95 confidence level.


**Rolling coverage to approximate conditional coverage**

Since conditional coverage is challenging to evaluate directly on real-world data, we instead approximate it using rolling coverage computed at each time index. We have added rolling coverage results on the wind dataset for our proposed method, as well as for the strong baseline methods that exhibited comparable coverage to FCP, in the appendix. These results show that our method consistently achieves conditional coverage close to the target coverage, with variances that are comparable to or smaller than those of the baseline methods.


**To improve Readability and Unified Notations**

We carefully reviewed the entire manuscript to unify notation and correct any confusing or inconsistent notations. Redundant parts were removed, and several sentences were polished to improve clarity and readability. We also moved the preliminary section into the main text for better flow. In addition, we added a visual overview of the method (Figure 1) and included an algorithmic summary (Algorithm 1) that outlines the overall training procedure.

---

### Meta-Review · Area_Chair_Cwx4 · 2026-01-12

**Summary:**

This paper proposes a flow-based conformal prediction framework for multi-dimensional time series that combines expressive generative modeling with conformal calibration to produce valid prediction sets in complex, structured temporal settings. The method extends conformal prediction beyond scalar or low-dimensional outputs by leveraging flow-based models to capture temporal and cross-dimensional dependencies, while preserving marginal coverage guarantees. The paper includes theoretical justification of validity and a comprehensive empirical evaluation on synthetic and real-world time series benchmarks.

While the reviews reflect some divergence in opinion, I find that the core contribution is technically sound, well motivated, and relevant to the ICLR audience. The paper addresses an important gap in conformal prediction for structured time series outputs, where existing methods are either overly simplistic or difficult to scale. The rebuttal effectively clarifies novelty relative to prior work, addresses misunderstandings about the role of flow-based modeling, and provides additional context for the empirical design choices. Overall, despite some remaining concerns about computational cost and assumptions, the contribution is sufficiently strong and timely to merit acceptance.

**Reviewer Concerns:**

Several reviewer concerns were addressed by the rebuttal. s5Xm initially raised strong concerns about novelty and relevance, but during the discussion explicitly acknowledged a misunderstanding regarding prior work and indicated that their assessment would change after clarification. Concerns raised by PND7 about theoretical assumptions and the scope of guarantees were addressed by clarifying that the primary goal is to ensure conformal validity while improving expressiveness of the conditional model, rather than to introduce new coverage notions. bXWm and W2dB raised questions about computational cost, sensitivity to model misspecification, and empirical comparisons, which were addressed through clearer explanation of experimental choices and limitations.

Some concerns remain partially outstanding. In particular, questions about scalability to very long or high-frequency time series and sensitivity to flow model hyperparameters remain open. However, these issues relate to scope and practical deployment rather than correctness or validity of the proposed method.

**Reviewer Scores:**

s5Xm (rating 0).
This reviewer explicitly stated during the discussion that their low rating was based on a misunderstanding of novelty and that they would update their score. I expect a substantial increase, likely to at least a borderline or positive rating.

PND7 (rating 4).
This reviewer expressed principled concerns about assumptions and scope. While the rebuttal clarified intent, I expect the rating would likely remain unchanged.

bXWm (rating 6).
This reviewer was mildly positive but comfortable with rejection. After discussion, the score would likely remain unchanged.

W2dB (rating 6).
This reviewer was positive on contribution but cautious about cost and sensitivity. The rebuttal addressed these points, but I expect the overall assessment would likely remain similar.

Overall, with clarification during discussion and at least one clear upward revision expected, the balance of evidence supports acceptance.

---

### Decision · Program_Chairs · 2026-01-26

Accept (Poster)